# PRACTICAL MARGINALIZED IMPORTANCE SAMPLING WITH THE SUCCESSOR REPRESENTATION

## ABSTRACT

Marginalized importance sampling (MIS), which measures the density ratio between the state-action occupancy of a target policy and that of a sampling distribution, is a promising approach for off-policy evaluation. However, current state-of-the-art MIS methods rely on complex optimization tricks and succeed mostly on simple toy problems. We bridge the gap between MIS and deep reinforcement learning by observing that the density ratio can be computed from the successor representation of the target policy. The successor representation can be trained through deep reinforcement learning methodology and decouples the reward optimization from the dynamics of the environment, making the resulting algorithm stable and applicable to high-dimensional domains. We evaluate the empirical performance of our approach on a variety of challenging Atari and MuJoCo environments.

## 1 INTRODUCTION

Off-policy evaluation (OPE) is a reinforcement learning (RL) task where the aim is to measure the performance of a target policy from data collected by a separate behavior policy (Sutton & Barto, 1998). As it can often be difficult or costly to obtain new data, OPE offers an avenue for re-using previously stored data, making it an important challenge for applying RL to real-world domains (Zhao et al., 2009; Mandel et al., 2014; Swaminathan et al., 2017; Gauci et al., 2018).

Marginalized importance sampling (MIS) (Liu et al., 2018; Xie et al., 2019; Nachum et al., 2019a) is a family of OPE methods which re-weight sampled rewards by directly learning the density ratio between the state-action occupancy of the target policy and the sampling distribution. This approach can have significantly lower variance than traditional importance sampling methods (Precup et al., 2001), which consider a product of ratios over trajectories, and is amenable to deterministic policies and behavior agnostic settings where the sampling distribution is unknown. However, the body of MIS work is largely theoretical, and as a result, empirical evaluations of MIS have mostly been carried out on simple low-dimensional tasks, such as mountain car (state dim. of 2) or cartpole (state dim. of 4). In comparison, deep RL algorithms have shown successful behaviors in high-dimensional domains such as Humanoid locomotion (state dim. of 376) and Atari (image-based).

In this paper, we present a straightforward approach for MIS that can be computed from the successor representation (SR) of the target policy. Our algorithm, the Successor Representation DIstribution Correction Estimation (SR-DICE), is the first method that allows MIS to scale to high-dimensional systems, far outperforming previous approaches. In comparison to previous algorithms which rely on minimax optimization or kernel methods (Liu et al., 2018; Nachum et al., 2019a; Uehara & Jiang, 2019; Mousavi et al., 2020), SR-DICE requires only a simple convex loss applied to the linear function determining the reward, after computing the SR. Similar to the deep RL methods which can learn in high-dimensional domains, the SR can be computed easily using behavior-agnostic temporal-difference (TD) methods. This makes our algorithm highly amenable to deep learning architectures and applicable to complex tasks.

Our derivation of SR-DICE also reveals an interesting connection between MIS methods and value function learning. The key motivation for MIS methods is, unlike traditional importance sampling methods, they can avoid variance with an exponential dependence on horizon, by re-weighting individual transitions rather than accumulating ratios along entire trajectories. We remark that while the MIS ratios only consider individual transitions, the optimization procedure is still subject to the

dynamics of the underlying MDP. Subsequently, we use this insight to show a connection between a well-known MIS method, DualDICE (Nachum et al., 2019a), and Bellman residual minimization (Bellman, 1957; Baird, 1995), which can help explain some of the optimization properties and performance of DualDICE, as well as other related MIS methods.

We benchmark the performance of SR-DICE on several high-dimensional domains in MuJoCo (Todorov et al., 2012) and Atari (Bellemare et al., 2013), against several recent MIS methods. Our results demonstrate two key findings regarding high-dimensional tasks.

**SR-DICE significantly outperforms the benchmark algorithms**. We attribute this performance gap to SR-DICE's deep RL components, outperforming the MIS baselines in the same way that deep RL outperforms traditional methods on high-dimensional domains. Unfortunately, part of this performance gap is due to the fact that the baseline MIS methods scale poorly to challenging tasks. In Atari we find that the baseline MIS method exhibit unstable estimates, often reaching errors with many orders of magnitude.

**MIS underperforms deep RL**. Although SR-DICE achieves a high performance, we find its errors are bounded by the quality of the SR. Consequently, we find that SR-DICE and the standard SR achieve a similar performance across all tasks. Worse so, we find that using a deep TD method, comparable to DQN (Mnih et al., 2015) for policy evaluation outperforms both methods. Although the performance gap is minimal, for OPE there lacks a convincing argument for SR-DICE, or any current MIS method, which introduce unnecessary complexity. However, this does not mean MIS is useless. We remark that the density ratios themselves are an independent objective which have been used for applications such as policy regularization (Nachum et al., 2019b; Touati et al., 2020), imitation learning (Kostrikov et al., 2019), off-policy policy gradients (Imani et al., 2018; Liu et al., 2019b; Zhang et al., 2019), and non-uniform sampling (Sinha et al., 2020). SR-DICE serves as a stable, scalable approach for computing these ratios. We provide extensive experimental details in the supplementary material and our code is made available.

## 2 BACKGROUND

**Reinforcement Learning.** RL is a framework for maximizing accumulated reward of an agent interacting with its environment (Sutton & Barto, 1998). This problem is typically framed as a Markov Decision Process (MDP) $(\mathcal{S}, \mathcal{A}, \mathcal{R}, p, d_0, \gamma)$, with state space $\mathcal{S}$, action space $\mathcal{A}$, reward function $\mathcal{R}$, dynamics model $p$, initial state distribution $d_0$ and discount factor $\gamma$. An agent selects actions according to a policy $\pi : \mathcal{S} \times \mathcal{A} \to [0, 1]$. In this paper we address the problem of off-policy evaluation (OPE) problem where the aim is to measure the normalized expected per-step reward of the policy $R(\pi) = (1 - \gamma)\mathbb{E}_\pi \left[ \sum_{t=0}^\infty \gamma^t r(s_t, a_t) \right]$. An important notion in OPE is the value function $Q^\pi(s, a) = \mathbb{E}_\pi[\sum_{t=0}^\infty \gamma^t r(s_t, a_t)|s_0 = s, a_0 = a]$, which measures the expected sum of discounted rewards when following $\pi$, starting from $(s, a)$.

We define $d^\pi(s, a)$ as the discounted state-action occupancy, the probability of seeing $(s, a)$ under policy $\pi$ with discount $\gamma$: $d^\pi(s, a) = (1 - \gamma)\sum_{t=0}^\infty \gamma^t \int_{s_0} d_0(s_0)p_\pi(s_0 \to s, t)\pi(a|s)d(s_0)$, where $p_\pi(s_0 \to s, t)$ is the probability of arriving at the state $s$ after $t$ time steps when starting from an initial state $s_0$. This distribution is important as $R(\pi)$ equals the expected reward $r(s, a)$ under $d^\pi$:

$$R(\pi) = \mathbb{E}_{(s,a)\sim d^\pi, r}[r(s, a)]. \tag{1}$$

**Successor Representation.** The successor representation (SR) (Dayan, 1993) of a policy is a measure of occupancy of future states. It can be viewed as a general value function that learns a vector of the expected discounted visitation for each state. The successor representation $\Psi^\pi$ of a given policy $\pi$ is defined as $\Psi^\pi(s'|s) = \mathbb{E}_\pi[\sum_{t=0}^\infty \gamma^t \mathbb{1}(s_t = s')|s_0 = s]$. Importantly, the value function can be recovered from the SR by summing over the expected reward of each state $V^\pi(s) = \sum_{s'} \Psi^\pi(s'|s)\mathbb{E}_{a'\sim\pi}[r(s', a')]$. For infinite state and action spaces, the SR can instead be generalized to the expected occupancy over features, known as the deep SR (Kulkarni et al., 2016) or successor features (Barreto et al., 2017). For a given encoding function $\phi : \mathcal{S} \times \mathcal{A} \to \mathbb{R}^n$, the deep SR $\psi^\pi : \mathcal{S} \times \mathcal{A} \to \mathbb{R}^n$ is defined as the expected discounted sum over features from the encoding function $\phi$ when starting from a given state-action pair and following $\pi$:

$$\psi^\pi(s, a) = \mathbb{E}_\pi \left[ \sum_{t=0}^\infty \gamma^t \phi(s_t, a_t) \bigg| s_0 = s, a_0 = a \right]. \tag{2}$$

If the encoding $\phi(s, a)$ is learned such that the original reward function is a linear function of the encoding $r(s, a) = \mathbf{w}^\top \phi(s, a)$, then similar to the original formulation of SR, the value function can be recovered from a linear function of the SR: $Q^\pi(s, a) = \mathbf{w}^\top \psi^\pi(s, a)$. The deep SR network $\psi^\pi$ is trained to minimize the MSE between $\psi^\pi(s, a)$ and $\phi(s, a) + \gamma \psi'(s', a')$ on transitions $(s, a, s')$ sampled from the data set. A frozen target network $\psi'$ is used to provide stability (Mnih et al., 2015; Kulkarni et al., 2016), and is updated to the current network $\psi' \leftarrow \psi^\pi$ after a fixed number of time steps. The encoding function $\phi$ is typically trained by an encoder-decoder network (Kulkarni et al., 2016; Machado et al., 2017; 2018a).

**Marginalized Importance Sampling.** Marginalized importance sampling (MIS) is a family of importance sampling approaches for off-policy evaluation in which the performance $R(\pi)$ is evaluated by re-weighting rewards sampled from a data set $\mathcal{D} = \{(s, a, r, s')\} \sim p(s'|s, a)d^\mathcal{D}(s, a)$, where $d^\mathcal{D}$ is an arbitrary distribution, typically but not necessarily, induced by some behavior policy. It follows that $R(\pi)$ can computed with importance sampling weights on the rewards $\frac{d^\pi(s,a)}{d^\mathcal{D}(s,a)}$:

$$R(\pi) = \mathbb{E}_{(s,a)\sim d^\mathcal{D}, r} \left[ \frac{d^\pi(s, a)}{d^\mathcal{D}(s, a)} r(s, a) \right]. \tag{3}$$

The goal of marginalized importance sampling methods is to learn the weights $w(s, a) \approx \frac{d^\pi(s,a)}{d^\mathcal{D}(s,a)}$, using data contained in $\mathcal{D}$. The main benefit of MIS is that unlike traditional importance methods, the ratios are applied to individual transitions rather than complete trajectories, which can reduce the variance of long or infinite horizon problems. In other cases, the ratios themselves can be used for a variety of applications which require estimating the occupancy of state-action pairs.

**DualDICE.** Dual stationary DIstribution Correction Estimation (DualDICE) (Nachum et al., 2019a) is a well-known MIS method which uses a minimax optimization to learn the density ratios. The underlying objective which DualDICE aims to minimize is the following:

$$\min_f J(f) := \frac{1}{2}\mathbb{E}_{(s,a)\sim d^\mathcal{D}} \left[ (f(s, a) - \gamma\mathbb{E}_{s',\pi}[f(s', a')])^2 \right] - (1 - \gamma)\mathbb{E}_{s_0,a_0\sim\pi}[f(s_0, a_0)]. \tag{4}$$

It can be shown that Equation (4) is uniquely optimized by the MIS density ratio. However, since $f(s, a) - \gamma\mathbb{E}_\pi[f(s', a')]$ is dependent on transitions $(s, a, s')$, there are two practical issues with this underlying objective. First, the objective contains a square within an expectation, giving rise to the double sampling problem (Baird, 1995), where the gradient will be biased when using only a single sample of $(s, a, s')$. Second, computing $f(s, a) - \gamma\mathbb{E}_{s',\pi}[f(s', a')]$ for arbitrary state-action pairs, particularly those not contained in the data set, is non-trivial, as it relies on an expectation over succeeding states, which is generally inaccessible without a model of the environment. To address both concerns, DualDICE uses Fenchel duality (Rockafellar, 1970) to create the following minimax optimization problem:

$$\min_f \max_w J(f, w) := \mathbb{E}_{(s,a)\sim d^\mathcal{D}, a'\sim\pi, s'} \left[ w(s, a)(f(s, a) - \gamma f(s', a')) - 0.5w(s, a)^2 \right] \\ - (1 - \gamma)\mathbb{E}_{s_0,a_0}[f(s_0, a_0)]. \tag{5}$$

Similar to the original formulation, Equation (4), it can be shown that Equation (5) is minimized when $w(s, a)$ is the desired density ratio.

## 3 A REWARD FUNCTION PERSPECTIVE ON DISTRIBUTION CORRECTIONS

In this section, we present our behavior-agnostic approach to estimating MIS ratios, called the Successor Representation DIstribution Correction Estimation (SR-DICE). Our main insight is that MIS can be viewed as an optimization over the reward function, where the loss is uniquely optimized when the reward is the desired density ratio. We then apply our reward function perspective on a well-known MIS method, DualDICE (Nachum et al., 2019a), which enables us to observe difficulties in the optimization process and better understand related methods. All proofs for this section are left to Appendix A.

### 3.1 THE SUCCESSOR REPRESENTATION DICE

We will now derive our MIS approach. Our derivation shows that by treating MIS as reward function optimization, we can obtain the desired density ratios can be obtained in a straightforward manner

from the SR of the target policy. This pushes the challenging aspect of learning onto the computation of the SR, rather than optimizing the density ratio estimate. Furthermore, when tackling high-dimensional tasks, we can leverage deep RL approaches (Mnih et al., 2015; Kulkarni et al., 2016) to make learning the SR stable, giving rise to a practical MIS method.

Our aim is to determine the MIS ratios $\frac{d^\pi(s,a)}{d^\mathcal{D}(s,a)}$, using only data sampled from the data set $\mathcal{D}$ and the policy $\pi$. This presents a challenge as we have direct access to neither $d^\pi$ nor $d^\mathcal{D}$. As a starting point, we begin by following the derivation of DualDICE (Nachum et al., 2019a). We first consider the convex function $\frac{1}{2}mx^2 - nx$, which is uniquely minimized by $x^* = \frac{n}{m}$. Now by replacing $x$ with $\hat{r}(s,a)$, $m$ with $d^\mathcal{D}(s,a)$, and $n$ with $d^\pi(s,a)$, we have reformulated the convex function as the following objective:

$$\min_{\hat{r}(s,a)\forall(s,a)} J(\hat{r}) := \frac{1}{2}\mathbb{E}_{(s,a)\sim d^\mathcal{D}}\left[\hat{r}(s,a)^2\right] - (1-\gamma)\mathbb{E}_{(s,a)\sim d^\pi}\left[\hat{r}(s,a)\right]. \tag{6}$$

While this objective is still impractical as it relies on expectations over both $d^\mathcal{D}$ and $d^\pi$, from Nachum et al. (2019a) we can state the following about Equation (6).

**Observation 1** *The objective $J(\hat{r})$ is minimized when $\hat{r}(s,a) = \frac{d^\pi(s,a)}{d^\mathcal{D}(s,a)}$, $\forall(s,a)$.*

Now we will diverge from the derivation of DualDICE. Note our choice of notation, $\hat{r}(s,a)$, in Equation (6). Describing the objective in terms of a fictitious reward $\hat{r}$ will allow us to draw on familiar relationships between rewards and value functions and build stronger intuition. Consider the equivalence between the value function over initial state-action pairs and the expectation of rewards over the state-action visitation of the policy $(1-\gamma)\mathbb{E}_{s_0,a_0}[Q^\pi(s_0,a_0)] = \mathbb{E}_{d^\pi}[r(s,a)]$. It follows that the expectation over $d^\pi$ in Equation (6) can be replaced with a value function $\hat{Q}^\pi$ over $\hat{r}$:

$$\min_{\hat{r}(s,a)\forall(s,a)} J(\hat{r}) := \frac{1}{2}\mathbb{E}_{(s,a)\sim d^\mathcal{D}}\left[\hat{r}(s,a)^2\right] - (1-\gamma)\mathbb{E}_{s_0,a_0}\left[\hat{Q}^\pi(s_0,a_0)\right]. \tag{7}$$

Using $(1-\gamma)\mathbb{E}_{s_0,a_0}\left[\hat{Q}^\pi(s_0,a_0)\right] = \mathbb{E}_{d^\pi}\left[\hat{r}(s,a)\right]$ provides a method for accessing the otherwise intractable $d^\pi$. This form of the objective is convenient because we can estimate the expectation over $d^\mathcal{D}$ by sampling from the data set and $Q^\pi$ can be computed using any policy evaluation method.

While we can estimate both terms in Equation (7) with relative ease, the optimization problem is not directly differentiable and would require re-learning the value function $\hat{Q}^\pi$ with every adjustment to the learned reward $\hat{r}$. Fortunately, there exists a straightforward paradigm which enables direct reward function optimization known as successor representation (SR).

Consider the relationship between the SR $\Psi^\pi$ of the target policy $\pi$ and its value function $\mathbb{E}_{s_0,a_0}[Q^\pi(s_0,a_0)] = \mathbb{E}_{s_0}[V^\pi(s_0)] = \mathbb{E}_{s_0}\left[\sum_s \Psi^\pi(s|s_0)\mathbb{E}_\pi[r(s,a)]\right]$ in the tabular setting. It follows that we can create an optimization problem over the reward function $\hat{r}$ from Equation (7):

$$\min_{\hat{r}(s,a)\forall(s,a)} J_\Psi(\hat{r}) := \frac{1}{2}\mathbb{E}_{(s,a)\sim d^\mathcal{D}}\left[\hat{r}(s,a)^2\right] - (1-\gamma)\mathbb{E}_{s_0}\left[\sum_s \Psi^\pi(s|s_0)\mathbb{E}_{a\sim\pi}\left[\hat{r}(s,a)\right]\right]. \tag{8}$$

This objective can be generalized to continuous states by considering the deep SR $\psi^\pi$ over features $\phi(s,a)$ and optimizing the weights of a linear function $\mathbf{w}$. In this instance, the estimated density ratio $\hat{r}(s,a)$ is determined by $\mathbf{w}^\top\phi(s,a)$ and we can optimize $\mathbf{w}$ by minimizing the following:

$$\min_{\mathbf{w}} J(\mathbf{w}) := \frac{1}{2}\mathbb{E}_{d^\mathcal{D}}\left[(\mathbf{w}^\top\phi(s,a))^2\right] - (1-\gamma)\mathbb{E}_{s_0,a_0\sim\pi}\left[\mathbf{w}^\top\psi^\pi(s_0,a_0)\right]. \tag{9}$$

Since this optimization problem is convex, it has a closed form solution. Define $\mathcal{D}_0$ as the set of start states contained in $\mathcal{D}$. The unique optimizer of Equation (9) is as follows:

$$\min_{\mathbf{w}} J(\mathbf{w}) = (1-\gamma)\frac{|\mathcal{D}|}{\sum_{(s,a)\in\mathcal{D}}\phi(s,a)\sum_i \phi_i(s,a)}\frac{1}{|\mathcal{D}_0|}\sum_{s_0\in\mathcal{D}_0}\pi(a_0|s_0)\psi^\pi(s_0,a_0), \tag{10}$$

where $\phi_i$ is the ith entry of the vector $\phi$. However, we may generally prefer iterative, gradient-based solutions for scalability. We call the combination of learning the deep SR followed by optimizing Equation (9) the Successor Representation stationary DIstribution Correction Estimation

---

**Algorithm 1** SR-DICE

---

1: At each time step sample mini-batch of $N$ transitions $(s, a, r, s')$ and start states $s_0$ from $\mathcal{D}$.
2: **for** $t = 1$ to $T_1$ **do**
3:     $\min_{\phi, D} \frac{1}{2}(D(\phi(s, a)) - (s, a))^2.$         # Encoding $\phi$ loss
4: **for** $t = 1$ to $T_2$ **do**
5:     $\min_{\psi^\pi} \frac{1}{2}(\phi(s, a) + \gamma\psi'(s', a') - \psi^\pi(s, a))^2.$      # Deep successor representation $\psi^\pi$ loss
6: **for** $t = 1$ to $T_3$ **do**
7:     $\min_{\mathbf{w}} \frac{1}{2}(\mathbf{w}^\top \phi(s, a))^2 - (1 - \gamma)\mathbf{w}^\top \psi^\pi(s_0, a_0).$     # Density ratio $\mathbf{w}$ loss (Equation (9))
8: **Output:** $R(\pi)$ estimate $|\mathcal{D}|^{-1} \sum_{(s, a, r) \in \mathcal{D}} \mathbf{w}^\top \phi(s, a) r(s, a).$

---

(SR-DICE). SR-DICE is split into three learning phases: (1) learning the encoding $\phi$, (2) learning the deep SR $\psi^\pi$, and (3) optimizing Equation (9). For the first two phases we follow standard practices from prior work (Kulkarni et al., 2016; Machado et al., 2018a), training the encoding $\phi$ via an encoder-decoder network to reconstruct the transition and training the deep SR $\psi^\pi$ using TD learning-style methods. We summarize SR-DICE in Algorithm 1. Additional implementation-level details can be found in Appendix D.

Although it is difficult to make any guarantees on the accuracy of an approximate $\psi^\pi$ trained with deep RL techniques, if we assume $\psi^\pi$ is exact, then we can show that SR-DICE learns the least squares estimator to the desired density ratio.

**Theorem 1** *Assuming* $(1 - \gamma)\mathbb{E}_{s_0, a_0}[\psi^\pi(s_0, a_0)] = \mathbb{E}_{(s, a) \sim d^\pi}[\phi(s, a)]$, *then the optimizer* $\mathbf{w}^*$ *of the objective* $J(\mathbf{w})$ *is the least squares estimator of* $\int_{\mathcal{S} \times \mathcal{A}} \left( \mathbf{w}^\top \phi(s, a) - \frac{d^\pi(s, a)}{d^\mathcal{D}(s, a)} \right)^2 d(s, a).$

Hence, the main sources of error in SR-DICE are learning the encoding $\phi$ and the deep SR $\psi^\pi$. Notably, both of these steps are independent of the main optimization problem of learning $\mathbf{w}$, as we have shifted the challenging aspects of density ratio estimation onto learning the deep SR. This leaves deep RL to do the heavy lifting. The remaining optimization problem, Equation (9), only involves directly updating the weights of a linear function, and unlike many other MIS methods, requires no tricky minimax optimization.

SR-DICE can also be applied to any pre-existing SR, or included into standard deep RL algorithms (Mnih et al., 2015; Lillicrap et al., 2015; Hessel et al., 2017; Fujimoto et al., 2018) by treating the encoding $\phi$ as an auxiliary reward. This provides an alternate form of policy evaluation through MIS, or a method to access density ratios between the target policy and the sampling distribution, with possible applications to exploration, policy regularization, or unbiased off-policy gradients (Liu et al., 2019b; Nachum et al., 2019b; Touati et al., 2020).

### 3.2 REWARD FUNCTIONS & MIS: A CASE STUDY ON DUALDICE

One of the main attractions for MIS methods is they use importance sampling ratios which re-weight individual transitions rather than entire trajectories. While independent of the length of trajectories collected by the behavior policy, we remark the optimization problem is *not* independent of the implicit horizon defined by the discount factor $\gamma$ and MIS methods are still subject to the dynamics of the underlying MDP. In SR-DICE we explicitly handle the dynamics of the MDP by learning the SR with TD learning methods. In this case study, we examine a well-known MIS method, DualDICE (Nachum et al., 2019a), and discuss how it propagates updates through the MDP by considering its relationship to residual algorithms which minimize the mean squared Bellman error (Baird, 1995). By viewing other MIS methods through the lens of reward function optimization, we can understand their connection to value-based methods, shedding light on their optimization properties and challenges.

Recall the underlying objective of DualDICE:

$$\min_f J(f) := \frac{1}{2}\mathbb{E}_{(s, a) \sim d^\mathcal{D}}\left[ (f(s, a) - \gamma\mathbb{E}_{s', \pi}[f(s', a')])^2 \right] - (1 - \gamma)\mathbb{E}_{s_0, a_0 \sim \pi}[f(s_0, a_0)]. \quad (11)$$

By viewing the problem as reward function optimization, we can transform DualDICE into a more familiar format that considers rewards and value functions. To begin, we state the following theorem.

**Theorem 2** *Given an MDP $(\mathcal{S}, \mathcal{A}, \cdot, p, d_0, \gamma)$, policy $\pi$, and function $f : \mathcal{S} \times \mathcal{A} \to \mathbb{R}$, define the reward function $r : \mathcal{S} \times \mathcal{A} \to \mathbb{R}$ where $\hat{r}(s, a) = f(s, a) - \gamma \mathbb{E}_{s',a' \sim \pi}[f(s', a')]$. Then it follows that the value function $\hat{Q}^\pi$ defined by the policy $\pi$, MDP, and reward function $\hat{r}$, is the function $f$.*

The proof follows naturally from the Bellman equation (Bellman, 1957). Informally, Theorem 2 states that any function $f$ can be treated as an exact value function $\hat{Q}^\pi$, for a carefully chosen reward function $\hat{r}(s, a) = f(s, a) - \gamma \mathbb{E}_{s',\pi}[f(s', a')]$.

Theorem 2 provides two perspectives on DualDICE. By replacing terms in Equation (11) with rewards and value functions, it can be viewed as the same objective as Equation (7) from SR-DICE:

$$\min_{\hat{r}(s,a) \forall (s,a)} J(\hat{r}) := \frac{1}{2} \mathbb{E}_{(s,a) \sim d^\mathcal{D}} \left[ \hat{r}(s, a)^2 \right] - (1 - \gamma) \mathbb{E}_{s_0, a_0} \left[ \hat{Q}^\pi(s_0, a_0) \right]. \tag{12}$$

The first insight from this relationship is that like SR-DICE, DualDICE can be viewed as reward function optimization and still requires some element of value learning. However, for DualDICE the form of the reward and value functions are unique. From Theorem 2, we remark that $f(s_0, a_0)$ is *always* exactly $\hat{Q}^\pi(s_0, a_0)$ without additional computation. This occurs because $f(s_0, a_0)$ is not a function of the reward, rather, the rewards are defined as a function of $f$. When the reward function is adjusted, $f(s_0, a_0)$ may remain unchanged and other rewards are adjusted to compensate.

To emphasize how DualDICE is subject to the properties of value learning, consider a second perspective on DualDICE taken from Theorem 2, where we replace $f$ with $\hat{Q}^\pi$:

$$\min_{\hat{Q}^\pi} J(\hat{Q}^\pi) := \underbrace{\frac{1}{2} \mathbb{E}_{(s,a) \sim d^\mathcal{D}} \left[ \left( \hat{Q}^\pi(s, a) - \gamma \mathbb{E}_{s',\pi} \left[ \hat{Q}^\pi(s', a') \right] \right)^2 \right]}_{\text{Bellman residual minimization}} - (1 - \gamma) \mathbb{E}_{s_0, a_0} \left[ \hat{Q}^\pi(s_0, a_0) \right]. \tag{13}$$

The first term is equivalent to Bellman residual minimization (Bellman, 1957; Baird, 1995), where the reward is 0 for all state-action pairs. The second term attempts to maximize only the initial value function $\hat{Q}^\pi(s_0, a_0)$. From a practical perspective this relationship is concerning as the first term relies on successfully propagating updates throughout the MDP to balance out with changes to the initial values, which may occur quickly. Consequently, in cases where DualDICE performs poorly, we may see the initial values approach infinity.

To understand how this objective performs empirically, we measured the output of DualDICE on a basic OPE task with an identical behavior and target policy. In this case the true MIS ratio is 1.0 for all state-action pairs. Consequently, both the fictiuous reward $\mathbb{E}_{d^\mathcal{D}}[f(s, a) - \gamma \mathbb{E}_{s',\pi}[f(s', a')]]$ and normalized initial value function $(1 - \gamma)\mathbb{E}_{s_0, a_0}[f(s_0, a_0)]$ should approach 1.0. In Figure 1, we graph both $\mathbb{E}_{d^\mathcal{D}}[w(s, a)]$, where $w(s, a) \approx f(s, a) - \gamma \mathbb{E}_{s',\pi}[f(s', a')]$ is the ratio used by DualDICE (Equation (5)), and $(1 - \gamma)\mathbb{E}_{s_0, a_0}[f(s_0, a_0)]$ output by DualDICE.

While on the easier task, Pendulum, the performance looks reasonable, on HalfCheetah we can see that $(1 - \gamma)\mathbb{E}_{s_0, a_0}[f(s_0, a_0)]$ greatly overestimates and $\mathbb{E}_{d^\mathcal{D}}[w(s, a)]$ is highly unstable. This result is intuitive given the form of

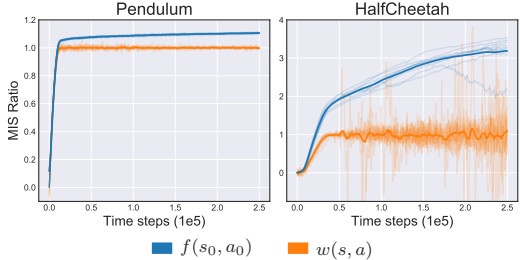

Figure 1: We plot the average values of $\mathbb{E}_{d^\mathcal{D}}[w(s, a)]$ and $(1 - \gamma)\mathbb{E}_{s_0, a_0}[f(s_0, a_0)]$ output by DualDICE on a task with an identical behavior and target policy, such that the true value of both terms is 1. The 10 individual trials are plotted lightly, with the mean in bold. The estimates of DualDICE matches our hypothesis that DualDICE overestimates $f(s_0, a_0)$ as propagating updates through the MDP occurs at a much slower rate.

Equation (13), where the first term, which $w(s, a)$ approximates, is pushed slowly towards 0 and the second term is pushed towards $\infty$. On the lower dimensional problem, Pendulum, the objective is optimized more easily and both terms approach 1.0. On the harder problem, HalfCheetah, we can see how balancing residual learning, which is notoriously slow (Baird, 1995), with a maximization term on initial states creates a difficult optimzation procedure.

These results highlight the importance, and challenge, of propagating updates through the MDP. MIS methods are not fundamentally different than value-based methods, and viewing them as such may allow us to develop richer foundations for MIS.

## 4 RELATED WORK

**Off-Policy Evaluation.** Off-policy evaluation is a well-studied problem with several families of approaches. One family of approaches is based on importance sampling, which re-weights trajectories by the ratio of likelihoods under the target and behavior policy (Precup et al., 2001). Importance sampling methods are unbiased but suffer from variance which can grow exponentially with the length of trajectories (Li et al., 2015; Jiang & Li, 2016). Consequently, research has focused on variance reduction (Thomas & Brunskill, 2016; Munos et al., 2016; Farajtabar et al., 2018) or contextual bandits (Dudík et al., 2011; Wang et al., 2017). Marginalized importance sampling methods (Liu et al., 2018) aim to avoid this exponential variance by considering the ratio in stationary distributions, giving an estimator with variance which is polynomial with respect to horizon (Xie et al., 2019; Liu et al., 2019a). Follow-up work has introduced a variety of approaches and improvements, allowing them to be behavior-agnostic (Nachum et al., 2019a; Uehara & Jiang, 2019; Mousavi et al., 2020) and operate in the undiscounted setting (Zhang et al., 2020a;c). In a similar vein, some OPE methods rely on emphasizing, or re-weighting, updates based on their stationary distribution (Sutton et al., 2016; Mahmood et al., 2017; Hallak & Mannor, 2017; Gelada & Bellemare, 2019), or learning the stationary distribution directly (Wang et al., 2007; 2008).

**Successor Representation.** Introduced originally by Dayan (1993) as an approach for improving generalization in temporal-difference methods, successor representations (SR) were revived by recent work on deep successor RL (Kulkarni et al., 2016) and successor features (Barreto et al., 2017) which demonstrated that the SR could be generalized to a function approximation setting. The SR has found applications for task transfer (Barreto et al., 2018; Grimm et al., 2019), navigation (Zhang et al., 2017; Zhu et al., 2017), and exploration (Machado et al., 2018a; Janz et al., 2019). It has also been used in a neuroscience context to model generalization and human reinforcement learning (Gershman et al., 2012; Momennejad et al., 2017; Gershman, 2018). The SR and our work also relate to state representation learning (Lesort et al., 2018) and general value functions (Sutton & Tanner, 2005; Sutton et al., 2011).

## 5 EXPERIMENTS

To evaluate our method, we perform several off-policy evaluation (OPE) experiments on a variety of domains. The aim is to evaluate the normalized average discounted reward $\mathbb{E}_{(s,a)\sim d^\pi,r}[r(s,a)]$ of a target policy $\pi$. We benchmark our algorithm against two MIS methods, DualDICE (Nachum et al., 2019a) and GradientDICE (Zhang et al., 2020c), two deep RL approaches and the true return of the behavior policy. The first deep RL method is a DQN-style approach (Mnih et al., 2015) where actions are selected by $\pi$ (denoted Deep TD) and the second is the deep SR where the weight $\mathbf{w}$ is trained to minimize the MSE between $\mathbf{w}^\top \phi(s,a)$ and $r(s,a)$ (denoted Direct-SR) (Kulkarni et al., 2016). Environment-specific experimental details are presented below and complete algorithmic and hyper-parameter details are included in the supplementary material.

**Continuous-Action Experiments.** We evaluate the methods on a variety of MuJoCo environments (Brockman et al., 2016; Todorov et al., 2012). We examine two experimental settings. In both settings the target policy $\pi$ and behavior policy $\pi_b$ are stochastic versions of a deterministic policy $\pi_d$ obtained from training the TD3 algorithm (Fujimoto et al., 2018). We evaluate a target policy $\pi = \pi_d + \mathcal{N}(0, \sigma^2)$, where $\sigma = 0.1$.

- For the "easy" setting, we gather a data set of 500k transitions using a behavior policy $\pi_b = \pi_d + \mathcal{N}(0, \sigma_b^2)$, where $\sigma_b = 0.133$. This setting roughly matches the experimental setting defined by Zhang et al. (2020a).
- For the "hard" setting, we gather a significantly smaller data set of 50k transitions using a behavior policy which acts randomly with $p = 0.2$ and uses $\pi_d + \mathcal{N}(0, \sigma_b^2)$, where $\sigma_b = 0.2$, with $p = 0.8$.

Unless specified otherwise, we use a discount factor of $\gamma = 0.99$ and all hyper-parameters are kept constant across environments. All experiments are performed over 10 seeds. We display the results of the "easy" setting in Figure 2 and the "hard" setting in Figure 3.

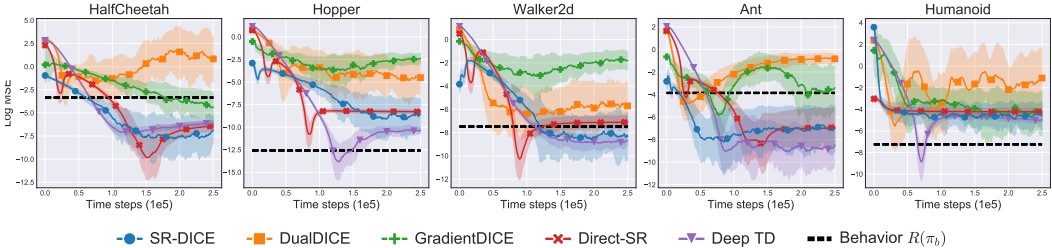

Figure 2: Off-policy evaluation results on the continuous-action MuJoCo domain using the "easy" experimental setting (500k time steps and $\sigma_b = 0.133$). The shaded area captures one standard deviation across 10 trials. We remark that this setting can be considered easy as the behavior policy achieves a lower error, often outperforming all agents. SR-DICE significantly outperforms the other MIS methods on all environments, except for Humanoid, where GradientDICE achieves a comparable performance.

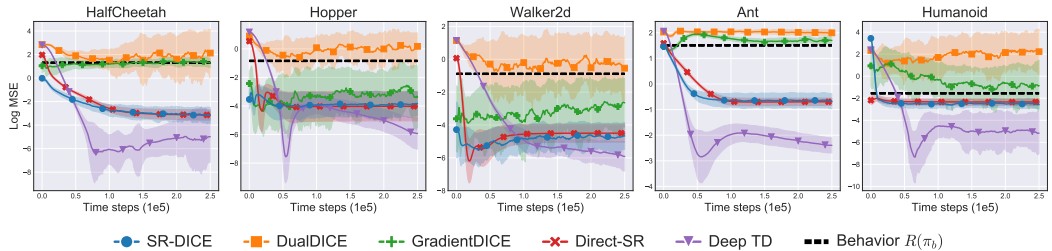

Figure 3: Off-policy evaluation results on the continuous-action MuJoCo domain using the "hard" experimental setting (50k time steps, $\sigma_b = 0.2$, random actions with $p = 0.2$). The shaded area captures one standard deviation across 10 trials. This setting uses significantly fewer time steps than the "easy" setting and the behavior policy is a poor estimate of the target policy. Again, we see SR-DICE outperforms the MIS methods, demonstrating the benefits of our proposed decomposition and simpler optimization. This setting also shows the benefits of deep RL methods over MIS methods for OPE in high dimensional domains, as deep TD performs the strongest in every environment.

**Atari Experiments.** We also test each method on several Atari games (Bellemare et al., 2013), which are challenging due to their high-dimensional image-based state space. Standard pre-processing steps are applied (Castro et al., 2018) and sticky-actions are used (Machado et al., 2018b) to increase difficulty and remove determinism. Each method is trained on a data set of one million time steps. The target policy is the deterministic greedy policy trained by Double DQN (Van Hasselt et al., 2016). The behavior policy is the $\epsilon$-greedy policy with $\epsilon = 0.1$. We use a discount factor of $\gamma = 0.99$. Experiments are performed over 3 seeds. Results are displayed in Figure 4. Additional experiments with different behavior policies can be found in the supplementary material.

**Discussion.** Across the board we find SR-DICE significantly outperforms the MIS methods. From the MSE graphs, we can see SR-DICE achieves much lower error in every task. Looking at the estimated values of $R(\pi)$ in the continuous-action environments, Figure 3, we can see that SR-DICE converges rapidly and maintains a stable estimate, while the MIS methods are particularly unstable, especially in the case of DualDICE. These observations are consistent in the Atari domain (Figure 4). Overall, we find the general trend in performance is Deep TD > SR-DICE = Direct-SR > MIS. Notably Direct-SR and SR-DICE perform similarly in every task, suggesting that the limiting factor in SR-DICE is the quality of the deep successor representation.

**Ablation.** To study the robustness of SR-DICE relative to the competing methods, we perform an ablation study and investigate the effects of data set size, discount factor, and two different behavior policies. Unless specified otherwise, we use experimental settings matching the "hard" setting. We report the results in Figure 5. In the data set size experiment (a), SR-DICE perform well with as few as 5k transitions (5 trajectories). In some instances, the performance is unexpectedly improved with less data, although incrementally. For small data sets, the SR methods outperform Deep TD. One hypothesis is that the encoding acts as an auxiliary reward and helps stabilize learning in the low data regime. In (b) we report the performance over changes in discount factor. The relative ordering across methods is unchanged. In (c) we use a behavior policy of $\mathcal{N}(0, \sigma_b^2)$, with $\sigma_b = 0.5$,

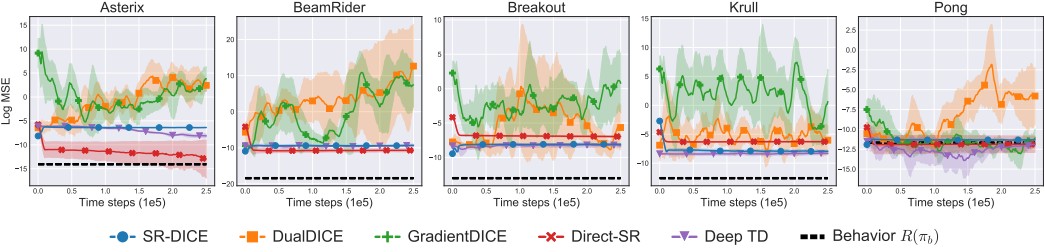

Figure 4: We plot the log MSE for off-policy evaluation in the image-based Atari domain. The shaded area captures one standard deviation across 3 trials. We can see the MIS baselines diverge on this challenging environment, while the remaining methods perform similarly. Perhaps surprisingly, on most games, the naïve baseline of using $R(\pi_b)$ from the behavior policy outperforms all methods by a fairly significant margin. Although the estimates from deep RL methods are stable, they are biased, resulting in a higher MSE.

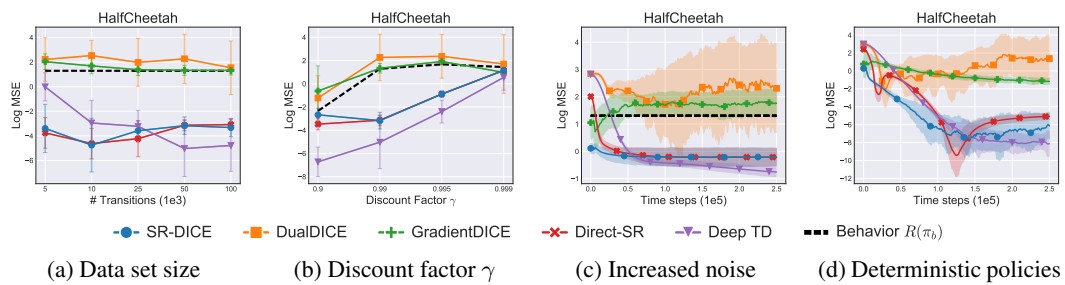

Figure 5: Ablation study results for the HalfCheetah task. We default to the "hard" setting wherever possible. Error bars and the shaded area captures one standard deviation over 10 trials. (a) We vary the size of the data set $\mathcal{D}$. (b) We vary the discount factor $\gamma$. (c) We use a new behavior policy with $\mathcal{N}(0, \sigma_b^2)$ noise with $\sigma_b = 0.5$. (d) We use the same deterministic behavior and target policy.

a much larger standard deviation than either setting for continuous control. The results are similar to the original setting, with an increased bias on the deep RL methods. In (d) we use the underlying deterministic policy as both the behavior and target policy. The baseline MIS methods perform surprisingly poorly, once again demonstrating their weakness on harder domains.

## 6 CONCLUSION

In this paper, we introduce a method which can perform marginalized importance sampling (MIS) using the successor representation (SR) of the target policy. This is achieved by deriving an MIS formulation that can be viewed as reward function optimization. By using the SR, we effectively disentangle the dynamics of the environment from learning the reward function. This allows us to (a) use well-known deep RL methods to effectively learn the SR in challenging domains (Mnih et al., 2015; Kulkarni et al., 2016) and (b) provide a straightforward loss function to learn the density ratios without any optimization tricks necessary for previous methods (Liu et al., 2018; Uehara & Jiang, 2019; Nachum et al., 2019a; Zhang et al., 2020c). This reward function interpretation also provides insight into prior MIS methods by showing how they are connected to value-based methods. Our resulting algorithm, SR-DICE, outperforms prior MIS methods in terms of both performance and stability and is the first MIS method which demonstrably scales to high-dimensional problems.

As a secondary finding, our benchmarking shows that current MIS methods underperform more traditional value-based methods at OPE on high-dimensional tasks, suggesting that for practical applications, deep RL approaches should still be preferred. Regardless, outside of OPE there exists a wealth of possible applications for MIS ratios, from imitation (Kostrikov et al., 2019) to policy optimization (Imani et al., 2018; Liu et al., 2019b; Zhang et al., 2019) to mitigating distributional shift in offline RL (Fujimoto et al., 2019b; Kumar et al., 2019). For ease of use, our code is provided, and we hope our algorithm and insight will provide valuable contributions to the field.

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

# A DETAILED PROOFS.

## A.1 OBSERVATION 1

**Observation 1** *The objective $J(\hat{r})$ is minimized when $\hat{r}(s,a) = \frac{d^\pi(s,a)}{d^\mathcal{D}(s,a)}$, $\forall(s,a)$.*

$$\min_{\hat{r}(s,a)\forall(s,a)} J(\hat{r}) := \frac{1}{2}\mathbb{E}_{(s,a)\sim d^\mathcal{D}}\left[\hat{r}(s,a)^2\right] - (1-\gamma)\mathbb{E}_{s_0,a_0}\left[\hat{Q}^\pi(s_0,a_0)\right] \qquad (14)$$

$$:= \frac{1}{2}\mathbb{E}_{(s,a)\sim d^\mathcal{D}}\left[\hat{r}(s,a)^2\right] - \mathbb{E}_{(s,a)\sim d^\pi}\left[\hat{r}(s,a)\right]. \qquad (15)$$

*Proof.*

Take the partial derivative of $J(\hat{r})$ with respect to $\hat{r}(s,a)$:

$$\frac{\partial}{\partial\hat{r}(s,a)}\left(\frac{1}{2}\mathbb{E}_{(s,a)\sim d^\mathcal{D}}\left[\hat{r}(s,a)^2\right] - \mathbb{E}_{(s,a)\sim d^\pi}\left[\hat{r}(s,a)\right]\right) = d^\mathcal{D}(s,a)\hat{r}(s,a) - d^\pi(s,a). \qquad (16)$$

Then setting $\frac{\partial J(\hat{r})}{\partial\hat{r}(s,a)} = 0$, we have that $J(\hat{r})$ is minimized when $\hat{r}(s,a) = \frac{d^\pi(s,a)}{d^\mathcal{D}(s,a)}$ for all state-action pairs $(s,a)$.

$\blacksquare$

## A.2 THEOREM 1

**Theorem 1** *Assuming $(1-\gamma)\mathbb{E}_{s_0,a_0}\left[\psi^\pi(s_0,a_0)\right] = \mathbb{E}_{(s,a)\sim d^\pi}[\phi(s,a)]$, then the optimizer $\mathbf{w}^*$ of the objective $J(\mathbf{w})$ is the least squares estimator of $\int_{\mathcal{S}\times\mathcal{A}}\left(\mathbf{w}^\top\phi(s,a) - \frac{d^\pi(s,a)}{d^\mathcal{D}(s,a)}\right)^2 d(s,a)$.*

$$\min_{\mathbf{w}} J(\mathbf{w}) := \frac{1}{2}\mathbb{E}_{d^\mathcal{D}}\left[(\mathbf{w}^\top\phi(s,a))^2\right] - (1-\gamma)\mathbb{E}_{s_0,a_0\sim\pi}\left[\mathbf{w}^\top\psi^\pi(s_0,a_0)\right]. \qquad (17)$$

*Proof.*

From our assumption we have:

$$\min_{\mathbf{w}} J(\mathbf{w}) := \frac{1}{2}\mathbb{E}_{d^\mathcal{D}}\left[(\mathbf{w}^\top\phi(s,a))^2\right] - \mathbb{E}_{(s,a)\sim d^\pi}\left[\mathbf{w}^\top\phi(s,a)\right]. \qquad (18)$$

Let $M = |\mathcal{S}| \times |\mathcal{A}|$ and $N$ be the feature dimension. Let $\phi(s,a)$ be a $N \times 1$ feature vector and $\Phi$ the $M \times N$ matrix where each row corresponds to a $\phi(s,a)^\top$ vector. Let $\mathbf{w}$ be a $N \times 1$ vector of parameters. Let $d_\pi$ and $d_\mathcal{D}$ be $M \times 1$ vectors of the values of $d^\pi(s,a)$ and $d^\mathcal{D}(s,a)$ for all $(s,a)$.

First note the least squares estimator of $\int_{\mathcal{S}\times\mathcal{A}}\left(\mathbf{w}^\top\phi(s,a) - \frac{d^\pi(s,a)}{d^\mathcal{D}(s,a)}\right)^2 d(s,a) = \left\|\Phi\mathbf{w} - \frac{d_\pi}{d_\mathcal{D}}\right\|^2$ is $\hat{\mathbf{w}} = (\Phi^T\Phi)^{-1}\Phi^T\frac{d^\pi}{d^\mathcal{D}}$, where the division is element-wise.

Now consider our optimization problem:

$$\begin{aligned}
J(\mathbf{w}) &= 0.5d_\mathcal{D}^\top(\Phi\mathbf{w})^2 - d_\pi^\top\Phi\mathbf{w} \\
&= 0.5d_\mathcal{D}^\top(\Phi\mathbf{w})^T\Phi\mathbf{w} - d_\pi^\top\Phi\mathbf{w} \qquad (19) \\
&= 0.5d_\mathcal{D}^\top\mathbf{w}^T\Phi^T\Phi\mathbf{w} - d_\pi^\top\Phi\mathbf{w}.
\end{aligned}$$

Now take gradient of $J(\mathbf{w})$ with respect to $\mathbf{w}$ and set it equal to 0:

$$
\begin{aligned}
\nabla_{\mathbf{w}} 0.5 d_{\mathcal{D}}^{\top} \mathbf{w}^T \Phi^T \Phi \mathbf{w} - d_{\pi}^{\top} \Phi \mathbf{w} &= 0 \\
d_{\mathcal{D}}^{\top} \mathbf{w}^T \Phi^T \Phi &= d_{\pi}^{\top} \Phi d^{\pi} \\
\mathbf{w}^T \Phi^T \Phi &= \left( \frac{d_{\pi}}{d_{\mathcal{D}}} \right)^{\top} \Phi \\
\Phi^T \Phi \mathbf{w} &= \Phi^{\top} \frac{d_{\pi}}{d_{\mathcal{D}}} \\
\mathbf{w} &= (\Phi^T \Phi)^{-1} \Phi^{\top} \frac{d_{\pi}}{d_{\mathcal{D}}}.
\end{aligned}
\tag{20}
$$

It follows that the optimizer to $J(\mathbf{w})$ is the least squares estimator.

∎

## A.3 THEOREM 2

**Theorem 2** *Given an MDP $(\mathcal{S}, \mathcal{A}, \cdot, p, d_0, \gamma)$, policy $\pi$, and function $f : \mathcal{S} \times \mathcal{A} \to \mathbb{R}$, define the reward function $r : \mathcal{S} \times \mathcal{A} \to \mathbb{R}$ as $\hat{r}(s, a) = f(s, a) - \gamma \mathbb{E}_{s', a' \sim \pi}[f(s', a')]$. Then it follows that the value function $\hat{Q}^{\pi}$ defined by the policy $\pi$, MDP, and reward function $\hat{r}$, is the function $f$.*

*Proof.*

Define $\hat{r}(s, a) = f(s, a) - \gamma \mathbb{E}_{\pi}[f(s', a')]$. Then note for all state-action pairs $(s, a)$ we have $f(s, a) = \hat{r}(s, a) + \gamma \mathbb{E}_{\pi}[f(s', a')] = f(s, a) - \gamma \mathbb{E}_{\pi}[f(s', a')] + \gamma \mathbb{E}_{\pi}[f(s', a')] = f(s, a)$, satisfying the Bellman equation. It follows that $f = \hat{Q}^{\pi}$ by the uniqueness of the value function (Bertsekas & Tsitsiklis, 1996).

This result can also be obtained by considering state-action reward shaping (Wiewiora et al., 2003), and treating $f$ as the potential function.

∎

## B TABULAR SR-DICE

The experimental performance of SR-DICE, in particular its reliance on the SR can be partially explained by examining its tabular counterpart. In fact, we can show that the value estimate derived from SR-DICE is exactly equal to a value estimate derived directly from the SR.

Recall the form of SR-DICE's objective in a tabular setting, as a function of the SR $\Psi^{\pi}$:

$$
\min_{\hat{r}(s, a) \forall (s, a)} J_{\Psi}(\hat{r}) := \frac{1}{2} \mathbb{E}_{(s, a) \sim d^{\mathcal{D}}} \left[ \hat{r}(s, a)^2 \right] - (1 - \gamma) \mathbb{E}_{s_0} \left[ \sum_s \Psi^{\pi}(s|s_0) \mathbb{E}_{a \sim \pi} \left[ \hat{r}(s, a) \right] \right]. \tag{21}
$$

Which has the following gradient:

$$
\nabla_{\hat{r}(s, a)} J_{\Psi}(\hat{r}) := d^{\mathcal{D}}(s, a) \hat{r}(s, a) - (1 - \gamma) \sum_{s_0} p(s_0) \Psi^{\pi}(s|s_0) \pi(a|s). \tag{22}
$$

We can compute this gradient from samples. Define $\mathcal{D}_0$ as the set of start states $s_0 \in \mathcal{D}$. It follows:

$$
\begin{aligned}
\nabla_{\hat{r}(s, a)} J_{\Psi}(\hat{r}) := \ & \frac{1}{|\mathcal{D}|} \left( \sum_{(s', a') \in \mathcal{D}} \mathbb{1}(s' = s, a' = a) \right) \hat{r}(s, a) \\
& - (1 - \gamma) \frac{1}{|\mathcal{D}_0|} \sum_{s_0 \in \mathcal{D}_0} \Psi^{\pi}(s|s_0) \pi(a|s).
\end{aligned}
\tag{23}
$$

Setting the above gradient to 0 and solving for $\hat{r}(s, a)$ we have the optimizer of $J_\Psi(\hat{r})$.

$$\hat{r}(s, a) = (1 - \gamma)\frac{1}{|\mathcal{D}_0|} \sum_{s_0 \in \mathcal{D}_0} \Psi^\pi(s|s_0)\pi(a|s)\frac{|\mathcal{D}|}{\sum_{(s',a') \in \mathcal{D}} \mathbb{1}(s' = s, a' = a)}. \tag{24}$$

Now consider the MIS equation for estimating the objective $R(\pi) = (1 - \gamma)\mathbb{E}_{s_0}[V^\pi(s_0)]$, where $\hat{r}$ is an estimate of $\frac{d^\pi(s,a)}{d^\mathcal{D}(s,a)}$:

$$\frac{1}{|\mathcal{D}|} \sum_{(s,a) \in \mathcal{D}} \frac{d^\pi(s, a)}{d^\mathcal{D}(s, a)} r(s, a). \tag{25}$$

For convenience, assume every state-action pair $(s, a)$ is contained at least once in $\mathcal{D}$. Although the result holds regardless, this assumption allows us to avoid some cumbersome details. Replace $\frac{d^\pi(s,a)}{d^\mathcal{D}(s,a)}$ with $\hat{r}$ in Equation (25) and expand and simplify:

$$\frac{1}{|\mathcal{D}|} \sum_{(s,a) \in \mathcal{D}} (1 - \gamma)\frac{1}{|\mathcal{D}_0|} \sum_{s_0 \in \mathcal{D}_0} \Psi^\pi(s|s_0)\pi(a|s)\frac{|\mathcal{D}|}{\sum_{(s',a') \in \mathcal{D}} \mathbb{1}(s' = s, a' = a)} r(s, a) \tag{26}$$

$$= (1 - \gamma)\frac{1}{|\mathcal{D}_0|} \sum_{s_0 \in \mathcal{D}_0} \sum_{(s,a) \in \mathcal{D}} \Psi^\pi(s|s_0)\pi(a|s)\frac{1}{\sum_{(s',a') \in \mathcal{D}} \mathbb{1}(s' = s, a' = a)} r(s, a) \tag{27}$$

$$= (1 - \gamma)\frac{1}{|\mathcal{D}_0|} \sum_{s_0 \in \mathcal{D}_0} \sum_{(s,a) \in \mathcal{S} \times \mathcal{A}} \Psi^\pi(s|s_0)\pi(a|s)r(s, a). \tag{28}$$

Noting that $\sum_{(s,a) \in \mathcal{S} \times \mathcal{A}} \Psi^\pi(s|s_0)\pi(a|s)r(s, a) = V^\pi(s_0)$ we can see that SR-DICE returns the same solution as the SR solution for estimating $R(\pi)$.

## C  ADDITIONAL EXPERIMENTS

In this section, we include additional experiments and visualizations, covering extra domains, additional ablation studies, run time experiments and additional behavior policies in the Atari domain.

### C.1  EXTRA CONTINUOUS DOMAINS

Although our focus is on high-dimensional domains, the environments, Pendulum and Reacher, have appeared in several related MIS papers (Nachum et al., 2019a; Zhang et al., 2020a). Therefore, we have included results for these domains in Figure 6. All experimental settings match the experiments in the main body, and are described fully in Appendix F.

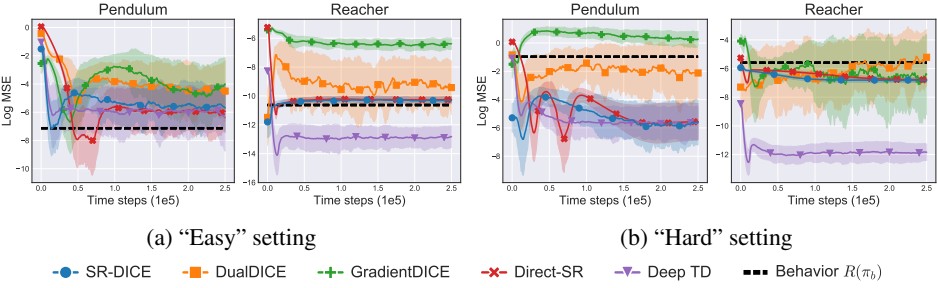

Figure 6: Off-policy evaluation results for Pendulum and Reacher. The shaded area captures one standard deviation across 10 trials. Even on these easier environment, we find that SR-DICE outperforms the baseline MIS methods.

## C.2 REPRESENTATION LEARNING & MIS

SR-DICE relies a disentangled representation learning phase where an encoding $\phi$ is learned, followed by the deep successor representation $\psi^\pi$ which are used with a linear vector $\mathbf{w}$ to estimate the density ratios. In this section we perform some experiments which attempt to evaluate the importance of representation learning by comparing their influence on the baseline MIS methods.

**Alternate representations.** We examine both DualDICE (Nachum et al., 2019a) and GradientDICE (Zhang et al., 2020c) under four settings where we pass the representations $\phi$ and $\psi^\pi$ to their networks, where both $\phi$ and $\psi^\pi$ are learned in identical fashion to SR-DICE.

(1) Input encoding $\phi$, $\qquad\qquad\qquad\qquad f(\phi(s,a)), \quad w(\phi(s,a)).$
(2) Input SR $\psi^\pi$, $\qquad\qquad\qquad\qquad\quad f(\psi^\pi(s,a)), \quad w(\psi^\pi(s,a)).$
(3) Input encoding $\phi$, linear networks, $\quad f^\top\phi(s,a), \quad w^\top\phi(s,a).$
(4) Input SR $\psi^\pi$, linear networks, $\qquad f^\top\psi^\pi(s,a), \quad w^\top\psi^\pi(s,a).$

See Appendix E for specific details on the baselines. We report the results in Figure 7. For GradientDICE, no benefit is provided by varying the representations, although using the encoding $\phi$ matches the performance of vanilla GradientDICE regardless of the choice of network, providing some validation that $\phi$ is a reasonable encoding. Interestingly, for DualDICE, we see performance gains from using the SR $\psi^\pi$ as a representation: slightly as input, but significantly when used with linear networks. On the other hand, as GradientDICE performs much worse with the SR, it is clear that the SR cannot be used as a representation without some degree of forethought.

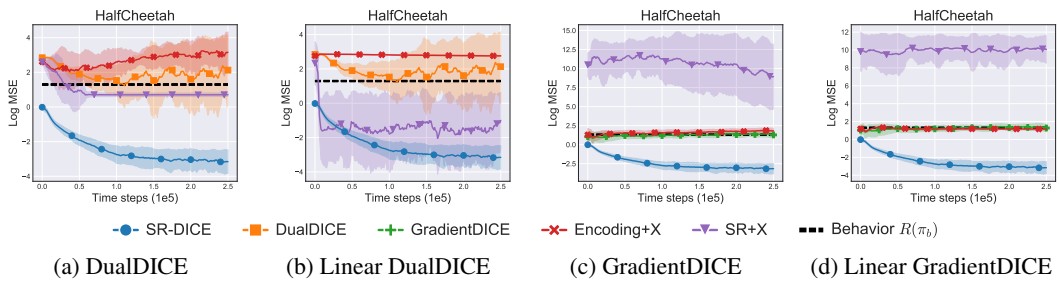

(a) DualDICE $\qquad$ (b) Linear DualDICE $\qquad$ (c) GradientDICE $\qquad$ (d) Linear GradientDICE

Figure 7: Off-policy evaluation results on HalfCheetah examining the value of differing representations added to the baseline MIS methods. The experimental setting corresponds to the "hard" setting from the main body. The shaded area captures one standard deviation across 10 trials. We see that using the SR $\psi^\pi$ as a representation improves the performance of DualDICE. On the other hand, GradientDICE performs much worse when using the SR, suggesting it cannot be used naively to improve MIS methods.

**Increased capacity.** As SR-DICE uses a linear function on top of a representation trained with the same capacity as the networks in DualDICE and GradientDICE, our next experiment examines if this additional capacity provides benefit to the baseline methods. To do, we expand each network in both baselines by adding an additional hidden layer. The results are reported in Figure 8. We find there is a very slight decrease in performance when using the larger capacity networks. This suggests the performance gap from SR-DICE over the baseline methods has little to do with model size.

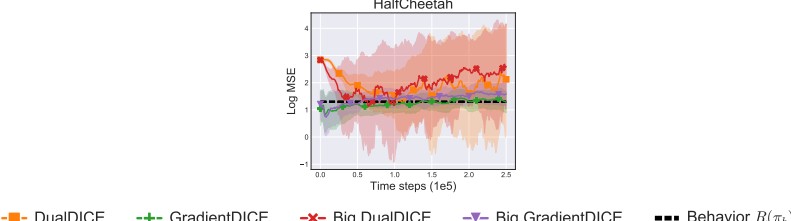

Figure 8: Off-policy evaluation results on HalfCheetah evaluating the performance benefits from larger network capacity on the baseline MIS methods. "Big" refers to the models with an additional hidden layer. The experimental setting corresponds to the "hard" setting from the main body. The shaded area captures one standard deviation across 10 trials. We find that there is no clear performance benefit from increasing network capacity.

### C.3 TOY DOMAINS

We additional test the MIS algorithms on a toy random-walk experiment with varying feature representations, based on a domain from (Sutton et al., 2009).

**Domain.** The domain is a simple 5-state MDP $(x_1, x_2, x_3, x_4, x_5)$ with two actions $(a_0, a_1)$, where action $a_0$ induces the transition $x_i \to x_{i-1}$ and action $a_1$ induces the transition $x_i \to x_{i+1}$, with the state $x_1$ looping to itself with action $a_0$ and $x_5$ looping to itself with action $a_5$. Episodes begin in the state $x_1$.

**Target.** We evaluate policy $\pi$ which selects actions uniformly, i.e. $\pi(a_0|x_i) = \pi(a_1|x_i) = 0.5$ for all states $x_i$. Our data set $\mathcal{D}$ contains all 10 possible state-action pairs and is sampled uniformly. We use a discount factor of $\gamma = 0.99$. Methods are evaluated on the average MSE between their estimate of $\frac{d^\pi}{d^{\mathcal{D}}}$ on all state-action pairs and the ground-truth value, which is calculated analytically.

**Hyper-parameters.** Since we are mainly interested in a function approximation setting, each method uses a small neural network with two hidden layers of 32, followed by tanh activation functions. All networks used stochastic gradient descent with a learning rate $\alpha$ tuned for each method out of $\{1, 0.5, 0.1, 0.05, 0.01, 0.001\}$. This resulted in $\alpha = 0.05$ for DualDICE, $\alpha = 0.1$ for GradientDICE, and $\alpha = 0.05$ for SR-DICE. Although there are a small number of possible data points, we use a batch size of 128 to resemble the regular training procedure. As recommended by the authors we use $\lambda = 1$ for GradientDICE (Zhang et al., 2020c), which was not tuned. For SR-DICE, we update the target network at every time step $\tau = 1$, which was not tuned.

Since there are only 10 possible state-action pairs, we use the closed form solution for the vector $\mathbf{w}$ (Equation (10)). Additionally, we skip the state representation phase of SR-DICE, instead learning the SR $\psi^\pi$ over the given representation of each state, such that the encoding $\phi = x$. This allows us to test SR-DICE to a variety of representations rather than using a learned encoding. Consequently, with these choices, SR-DICE has no pre-training phase, and therefore, unlike every other graph in this paper, we report the results as the SR is trained, rather than as the vector $\mathbf{w}$ is trained.

**Features.** To test the robustness of each method we examine three versions of the toy domain, each using a different feature representation over the same 5-state MDP. These feature sets are again taken from (Sutton et al., 2009).

- Tabular features: states are represented by a one-hot encoding, for example $x_2 = [0, 1, 0, 0, 0]$.
- Inverted features: states are represented by the inverse of a one-hot encoding, for example $x_2 = \left[\frac{1}{2}, 0, \frac{1}{2}, \frac{1}{2}, \frac{1}{2}\right]$.
- Dependent features: states are represented by 3 features which is not sufficient to cover all states exactly. In this case $x_1 = [1, 0, 0]$, $x_2 = [\frac{1}{\sqrt{2}}, \frac{1}{\sqrt{2}}, 0]$, $x_3 = [\frac{1}{\sqrt{3}}, \frac{1}{\sqrt{3}}, \frac{1}{\sqrt{3}}]$, $x_4 = [0, \frac{1}{\sqrt{2}}, \frac{1}{\sqrt{2}}]$, $x_5 = [0, 0, 1]$. Since our experiments use neural networks rather than linear functions, this representation is mainly meant to test SR-DICE, where we skip the state representation phase for SR-DICE and use the encoding $\phi = x$, limiting the representation of the SR.

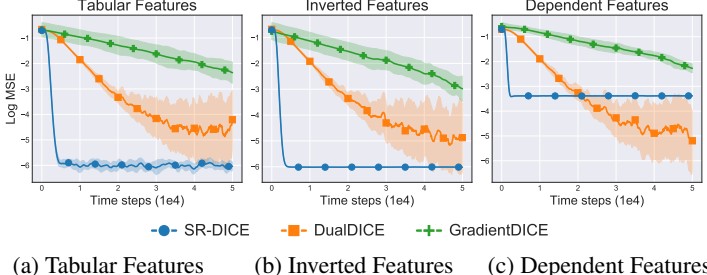

(a) Tabular Features     (b) Inverted Features     (c) Dependent Features

Figure 9: Results measuring the log MSE between the estimated density ratio and the ground-truth on a simple 5-state MDP domain with three feature sets. The shaded area captures one standard deviation across 10 trials. Results are evaluated every 100 time steps over 50k time steps total.

**Results.** We report the results in Figure 9. We remark on several observations. SR-DICE learns significantly faster than the baseline methods, likely due to its use of temporal difference methods in the SR update, rather than using an update similar to residual learning, which is notoriously

slow (Baird, 1995; Zhang et al., 2020b). GradientDICE appears to still be improving, although we limit training at 50k time steps, which we feel is sufficient given the domain is deterministic and only has 5 states. Notably, GradientDICE also uses a higher learning rate than SR-DICE and DualDICE. We also find the final performance of SR-DICE is much better than DualDICE and GradientDICE in the domains where the feature representation is not particularly destructive, highlighting the easier optimization of SR-DICE. In the case of the dependent features, we find DualDICE outperforms SR-DICE after sufficient updates. However, we remark that this concern could likely be resolved by learning the features and that SR-DICE still outperforms GradientDICE. Overall, we believe these results demonstrate that SR-DICE's strong empirical performance is consistent across simpler domains as well as the high dimensional domains we examine in the main body.

## C.4 RUN TIME EXPERIMENTS

In this section, we evaluate the run time of each algorithm used in our experiments. Although SR-DICE relies on pre-training the deep successor representation before learning the density ratios, we find each marginalized importance sampling (MIS) method uses a similar amount of compute, due to the reduced cost of training $\mathbf{w}$ after the pre-training phase.

We evaluate the run time on the HalfCheetah environment in MuJoCo (Todorov et al., 2012) and OpenAI gym (Brockman et al., 2016). As in the main set of experiments, each method is trained for 250k time steps. Additionally, SR-DICE and Direct-SR train the encoder-decoder for 30k time steps and the deep successor representation for 100k time steps before training $\mathbf{w}$. Run time is averaged over 3 seeds. All time-based experiments are run on a single GeForce GTX 1080 GPU and a Intel Core i7-6700K CPU. Results are reported in Figure 10.

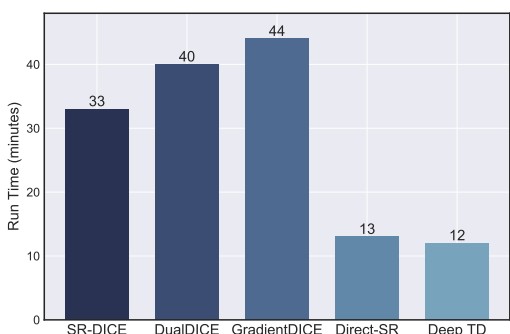

Figure 10: The average run time of each off-policy evaluation approach in minutes. Each experiment is run for 250k time steps and is averaged over 3 seeds. SR-DICE and Direct-SR pre-train encoder-decoder for 30k time steps and the deep successor representation 100k time steps.

We find the MIS algorithms run in a comparable time, regardless of the pre-training step involved in SR-DICE. This can be explained as training $\mathbf{w}$ in SR-DICE involves significantly less compute than DualDICE and GradientDICE which update multiple networks. On the other hand, the deep reinforcement learning approaches run in about half the time of SR-DICE.

## C.5 ATARI EXPERIMENTS

To better evaluate the algorithms in the Atari domain, we run two additional experiments where we swap the behavior policy. We observe similar trends as the experiments in the main body of the paper. In both experiments we keep all other settings fixed. Notably, we continue to use the same target policy, corresponding to the greedy policy trained by Double DQN (Van Hasselt et al., 2016), the same discount factor $\gamma = 0.99$, and the same data set size of 1 million.

**Increased noise.** In our first experiment, we attempt to increase the randomness of the behavior policy. As this can cause destructive behavior in the performance of the agent, we adopt an episode-dependent policy which selects between the noisy policy or the deterministic greedy policy at the beginning of each episode. This is motivated by the offline deep reinforcement learning experiments from (Fujimoto et al., 2019a). As a result, we use an $\epsilon$-greedy policy with $p = 0.8$ and the deter-

ministic greedy policy (the target policy) with $p = 0.2$. $\epsilon$ is set to $0.2$, rather than $0.1$ as in the experiments in the main body of the paper. Results are reported in Figure 11.

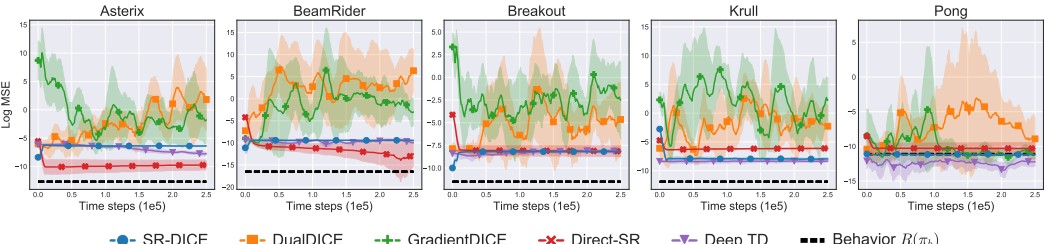

Figure 11: We plot the log MSE for off-policy evaluation in the image-based Atari domain, using an episode-dependent noisy policy, where $\epsilon = 0.2$ with $p = 0.8$ and $\epsilon = 0$ with $p = 0.2$. This episode-dependent selection ensures sufficient state-coverage while using a stochastic policy. The shaded area captures one standard deviation across 3 trials. Markers are not placed at every point for visual clarity.

We observe very similar trends to the original set of experiments. Again, we note DualDICE and GradientDICE perform very poorly, while SR-DICE, Direct-SR, and Deep TD achieve a reasonable, but biased, performance. In this setting, we still find the behavior policy is the closest estimate of the true value of $R(\pi)$ .

**Separate behavior policy.** In this experiment, we use a behavior which is distinct from the target policy, rather than simply adding noise. This behavior policy is derived from an agent trained with prioritized experience replay and Double DQN (Schaul et al., 2016). Again, we use a $\epsilon$-greedy policy, with $\epsilon = 0.1$. We report the results in Figure 12.

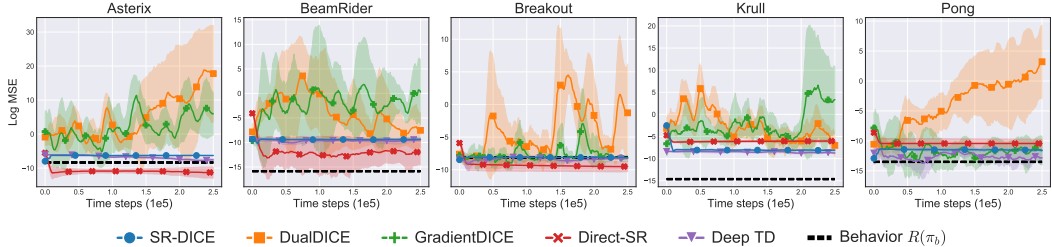

Figure 12: We plot the log MSE for off-policy evaluation in the image-based Atari domain, using a distinct behavior policy, trained by a separate algorithm, from the target policy. This experiment tests the ability to generalize to a more off-policy setting. The shaded area captures one standard deviation across 3 trials. Markers are not placed at every point for visual clarity.

Again, we observe similar trends in performance. Notably, in the Asterix game, the performance of Direct-SR surpasses the behavior policy, suggesting off-policy evaluation can outperform the naïve estimator in settings where the policy is sufficiently "off-policy" and distinct.

## D  SR-DICE PRACTICAL DETAILS

In this section, we cover some basic implementation-level details of SR-DICE. Note that code is provided for additional clarity.

SR-DICE uses two parametric networks, an encoder-decoder network to learn the encoding $\phi$ and a deep successor representation network $\psi^\pi$. Additionally, SR-DICE uses the weights of a linear function $\mathbf{w}$. SR-DICE begins by pre-training the encoder-decoder network and the deep successor representation before applying updates to $\mathbf{w}$.

**Encoder-Decoder.** This encoder-decoder network encodes $(s, a)$ to the feature vector $\phi(s, a)$, which is then decoded by several decoder heads. For the Atari domain, we choose to condition the feature vector only on states $\phi(s)$, as the reward is generally independent of the action selection. This

---

**Algorithm 2** SR-DICE

---

1: **Input:** Data set $\mathcal{D}$, target policy $\pi$, number of iterations $T_1$, $T_2$, $T_3$, mini-batch size $N$, target_update_rate.

---

2: **for** $t = 1$ **to** $T_1$ **do**
3:    Sample mini-batch of $N$ transitions $(s, a, r, s')$ from $\mathcal{D}$.
4:    $\min_{\phi, D_{s'}, D_a, D_r} \lambda_{s'}(D_{s'}(\phi(s, a)) - s')^2$                  Train encoder-
      $+\lambda_a(D_a(\phi(s, a)) - a)^2 + \lambda_r(D_r(\phi(s, a)) - r)^2.$       decoder
5: **end for**

---

6: **for** $t = 1$ **to** $T_2$ **do**
7:    Sample mini-batch of $N$ transitions $(s, a, r, s')$ from $\mathcal{D}$.
8:    Sample $a' \sim \pi(s')$.                              Train deep
9:    $\min_{\psi^\pi}(\phi(s, a) + \gamma\psi'(s', a') - \psi^\pi(s, a))^2.$       successor
10:   If $t$ mod target_update_rate $= 0$: $\psi' \leftarrow \psi$.     representation
11: **end for**

---

12: **for** $t = 1$ **to** $T_3$ **do**
13:   Sample mini-batch of $N$ transitions $(s, a, r, s')$ from $\mathcal{D}$.
14:   Sample mini-batch of $N$ start states $s_0$ from $\mathcal{D}$.
15:   Sample $a_0 \sim \pi(s_0)$.                      Learn $\mathbf{w}$
16:   $\min_{\mathbf{w}} \frac{1}{2}(\mathbf{w}^\top \phi(s, a))^2 - (1 - \gamma)\mathbf{w}^\top \psi^\pi(s_0, a_0).$
17: **end for**

---

change applies to both SR-DICE and Direct-SR. Most design decisions are inspired by prior work (Machado et al., 2017; 2018a).

For continuous control, given a mini-batch transition $(s, a, r, s')$, the encoder-decoder network is trained to map the state-action pair $(s, a)$ to the next state $s'$, the action $a$ and reward $r$. The resulting loss function is as follows:

$$\min_{\phi, D_{s'}, D_a, D_r} \mathcal{L}(\phi, D) := \lambda_{s'}(D_{s'}(\phi(s, a)) - s')^2 + \lambda_a(D_a(\phi(s, a)) - a)^2 + \lambda_r(D_r(\phi(s, a)) - r)^2.$$
(29)

We use $\lambda_{s'} = 1$, $\lambda_a = 1$ and $\lambda_r = 0.1$.

For the Atari games, given a mini-batch transition $(s, a, r, s')$, the encoder-decoder network is trained to map the state $s$ to the next state $s'$ and reward $r$, while penalizing the size of $\phi(s)$. The resulting loss function is as follows:

$$\min_{\phi, D_{s'}, D_r} \mathcal{L}(\phi, D) := \lambda_{s'}(D_{s'}(\phi(s)) - s')^2 + \lambda_r(D_r(\phi(s)) - r)^2 + \lambda_\phi \phi(s)^2.$$
(30)

We use $\lambda_{s'} = 1$, $\lambda_r = 0.1$ and $\lambda_\phi = 0.1$.

**Deep Successor Representation.** The deep successor representation $\psi^\pi$ is trained to estimate the accumulation of $\phi$. The training procedure resembles standard deep reinforcement learning algorithms. Given a mini-batch of transitions $(s, a, r, s')$ the network is trained to minimize the following loss:

$$\min_{\psi^\pi} \mathcal{L}(\psi^\pi) := (\phi(s, a) + \gamma\psi'(s', a') - \psi^\pi(s, a))^2,$$
(31)

where $\psi'$ is the target network. A target network is a frozen network used to provide stability (Mnih et al., 2015; Kulkarni et al., 2016) in the learning target. The target network is updated to the current network $\psi' \leftarrow \psi^\pi$ after a fixed number of time steps, or updated with slowly at each time step $\psi' \leftarrow \tau\psi^\pi + (1 - \tau)\psi^\pi$ (Lillicrap et al., 2015).

**Marginalized Importance Sampling Weights.** As described in the main body, we learn $\mathbf{w}$ by optimizing the following objective:

$$\min_{\mathbf{w}} J(\mathbf{w}) := \frac{1}{2}\mathbb{E}_{(s, a) \sim d^{\mathcal{D}}}\left[(\mathbf{w}^\top \phi(s, a))^2\right] - (1 - \gamma)\mathbb{E}_{s_0, a_0 \sim \pi}\left[\mathbf{w}^\top \psi^\pi(s_0, a_0)\right].$$
(32)

This is achieved by sampling state-action pairs uniformly from the data set $\mathcal{D}$, alongside a mini-batch of start states $s_0$, which are recorded at the beginning of each episode during data collection.

We summarize the learning procedure of SR-DICE in Algorithm 2.

# E    BASELINES

In this section, we cover some of the practical details of each of the baseline methods.

## E.1    DUALDICE

Dual stationary DIstribution Correction Estimation (DualDICE) (Nachum et al., 2019a) uses two networks $f$ and $w$. The general optimization problem is defined as follows:

$$\min_f \max_w J(f, w) := \mathbb{E}_{(s,a) \sim d^{\mathcal{D}}, a' \sim \pi, s'} \left[ w(s, a)(f(s, a) - \gamma f(s', a')) - 0.5 w(s, a)^2 \right]$$
$$- (1 - \gamma) \mathbb{E}_{s_0, a_0}[f(s_0, a_0)].$$
$$(33)$$

In practice this corresponds to alternating single gradient updates to $f$ and $w$. The authors suggest possible alternative functions to the convex function $0.5 w(s, a)^2$ such as $\frac{2}{3}|w(s, a)|^{\frac{3}{2}}$, however in practice we found $0.5 w(s, a)^2$ performed the best.

## E.2    GRADIENTDICE

Gradient stationary DIstribution Correction Estimation (GradientDICE) (Zhang et al., 2020c) uses two networks $f$ and $w$, and a scalar $u$. The general optimization problem is defined as follows:

$$\min_w \max_{f,u} J(w, u, f) := (1 - \gamma) \mathbb{E}_{s_0, a_0}[f(s_0, a_0)] + \gamma \mathbb{E}_{(s,a) \sim d^{\mathcal{D}}, a' \sim \pi, s'}[w(s, a) f(s', a')]$$
$$- \mathbb{E}_{(s,a) \sim d^{\mathcal{D}}}[w(s, a) f(s, a)] + \lambda \left( \mathbb{E}_{(s,a) \sim d^{\mathcal{D}}}[uw(s, a) - u] - 0.5 u^2 \right).$$
$$(34)$$

Similarly to DualDICE, in practice this involves alternating single gradient updates to $w$, $u$ and $f$. As suggested by the authors we use $\lambda = 1$.

## E.3    DIRECT-SR

Direct-SR is a policy evaluation version of deep successor representation (Kulkarni et al., 2016). The encoder-decoder network and deep successor representation are trained in the exact same manner as SR-DICE (see Section D). Then, rather than train $\mathbf{w}$ to learn the marginalized importance sampling ratios, $\mathbf{w}$ is trained to recover the original reward function. Given a mini-batch of transitions $(s, a, r, s')$, the following loss is applied:

$$\min_{\mathbf{w}} \mathcal{L}(\mathbf{w}) := (r - \mathbf{w}^\top \phi(s, a))^2.$$
$$(35)$$

## E.4    DEEP TD

Deep TD, short for deep temporal-difference learning, takes the standard deep reinforcement learning methodology, akin to DQN (Mnih et al., 2015), and applies it to off-policy evaluation. Given a mini-batch of transitions $(s, a, r, s')$ the Q-network is updated by the following loss:

$$\min_{Q^\pi} \mathcal{L}(Q^\pi) := (r + \gamma Q'(s', a') - Q^\pi(s, a))^2,$$
$$(36)$$

where $a'$ is sampled from the target policy $\pi(\cdot|s')$. Similarly, to training the deep successor representation, $Q'$ is a frozen target network which is updated to the current network after a fixed number of time steps, or incrementally at every time step.

# F    EXPERIMENTAL DETAILS

All networks are trained with PyTorch (version 1.4.0) (Paszke et al., 2019). Any unspecified hyperparameter uses the PyTorch default setting.

**Evaluation.** The marginalized importance sampling methods are measured by the average weighted reward from transitions sampled from a replay buffer $\frac{1}{N} \sum_{(s,a,r)} w(s,a)r(s,a)$, with $N = 10k$, while the deep RL methods use $\frac{(1-\gamma)}{M} \sum_{s_0} Q(s_0, \pi(a_0))$, where $M$ is the number of episodes. Each OPE method is trained on data collected by some behavioral policy $\pi_b$. We estimate the "true" normalized average discounted reward of the target and behavior policies from 100 roll-outs in the environment.

### F.1 CONTINUOUS-ACTION ENVIRONMENTS

Our agents are evaluated via tasks interfaced through OpenAI gym (version 0.17.2) (Brockman et al., 2016), which mainly rely on the MuJoCo simulator (mujoco-py version 1.50.1.68) (Todorov et al., 2012). We provide a description of each environment in Table 1.

Table 1: Continuous-action environment descriptions.

| Environment | State dim. | Action dim. | Episode Horizon | Task description |
|---|---|---|---|---|
| Pendulum-v0 | 3 | 1 | 200 | Balance a pendulum. |
| Reacher-v2 | 11 | 2 | 50 | Move end effector to goal. |
| HalfCheetah-v3 | 17 | 6 | 1000 | Locomotion. |
| Hopper-v3 | 11 | 3 | 1000 | Locomotion. |
| Walker2d-v3 | 17 | 6 | 1000 | Locomotion. |
| Ant-v3 | 111 | 8 | 1000 | Locomotion. |
| Humanoid-v3 | 376 | 17 | 1000 | Locomotion. |

**Experiments.** Our experiments are framed as off-policy evaluation tasks in which agents aim to evaluate $R(\pi) = \mathbb{E}_{(s,a) \sim d^{\pi}, r}[r(s,a)]$ for some target policy $\pi$. In each of our experiments, $\pi$ corresponds to a noisy version of a policy trained by a TD3 agent (Fujimoto et al., 2018), a commonly used deep reinforcement learning algorithm. Denote $\pi_d$, the deterministic policy trained by TD3 using the author's GitHub `https://github.com/sfujim/TD3`. The target policy is defined as: $\pi + \mathcal{N}(0, \sigma^2)$, where $\sigma = 0.1$. The off-policy evaluation algorithms are trained on a data set generated by a single behavior policy $\pi_b$. The experiments are done with two settings "easy" and "hard" which vary the behavior policy and the size of the data set. All other settings are kept fixed. For the "easy" setting the behavior policy is defined as:

$$\pi_b = \pi_d + \mathcal{N}(0, \sigma_b^2), \sigma_b = 0.133, \tag{37}$$

and 500k time steps are collected (approximately 500 trajectories for most tasks). The "easy" setting is roughly based on the experimental setting from Zhang et al. (2020a). For the "hard" setting the behavior policy adds an increased noise and selects random actions with $p = 0.2$:

$$\pi_b = \begin{cases} \pi_d + \mathcal{N}(0, \sigma_b^2), \sigma_b = 0.2 & p = 0.8, \\ \text{Uniform random action} & p = 0.2, \end{cases} \tag{38}$$

and only 50k time steps are collected (approximately 50 trajectories for most tasks). For Pendulum-v0 and Humanoid-v3, the range of actions is $[-2, 2]$ and $[-0.4, 0.4]$ respectively, rather than $[-1, 1]$, so we scale the size of the noise added to actions accordingly. We set the discount factor to $\gamma = 0.99$. All continuous-action experiments are over 10 seeds.

**Pre-training.** Both SR-DICE and Direct-SR rely on pre-training the encoder-decoder and deep successor representation $\psi$. These networks were trained for 30k and 100k time steps respectively. As noted in Section C.4, even when including this pre-training step, both algorithm have a lower running time than DualDICE and GradientDICE.

**Architecture.** For fair comparison, we use the same architecture for all algorithms except for DualDICE. This a fully connected neural network with 2 hidden layers of 256 and ReLU activation functions. This architecture was based on the network defined in the TD3 GitHub and was not tuned. For DualDICE, we found tanh activation functions improved stability over ReLU.

For SR-DICE and SR-Direct we use a separate architecture for the encoder-decoder network. The encoder is a network with a single hidden layer of 256, making each $\phi(s, a)$ a feature vector of 256.

There are three decoders for reward, action, and next state, respectively. For the action decoder and next state decoder we use a network with one hidden layer of 256. The reward decoder is a linear function of the encoding, without biases. All hidden layers are followed by ReLU activation functions.

**Network hyper-parameters.** All networks are trained with the Adam optimizer (Kingma & Ba, 2014). We use a learning rate of $3e-4$, again based on TD3 for all networks except for GradientDICE, which we found required careful tuning to achieve a reasonable performance. For GradientDICE we found a learning rate of $1e-5$ for $f$ and $w$, and $1e-2$ for $u$ achieved the highest performance. For DualDICE we chose the best performing learning rate out of $\{1e-2, 1e-3, 3e-4, 5e-5, 1e-5\}$. SR-DICE, Direct-SR and Deep TD were not tuned and use default hyper-parameters from deep RL algorithms. For training $\psi^\pi$ and $Q^\pi$ for the deep reinforcement learning aspects of SR-DICE, Direct-SR and Deep TD we use a mini-batch size of 256 and update the target networks using $\tau = 0.005$, again based on TD3. For all MIS methods, we use a mini-batch size of 2048 as described by (Nachum et al., 2019a). We found SR-DICE and DualDICE succeeded with lower mini-batch sizes but did not test this in detail. All hyper-parameters are described in Table 2.

Table 2: Continuous-action environment training hyper-parameters.

| Hyper-parameter | SR-DICE | DualDICE | GradientDICE | Direct-SR | Deep TD |
|---|---|---|---|---|---|
| Optimizer | Adam | Adam | Adam | Adam | Adam |
| $\psi^\pi$, $Q^\pi$ Learning rate | $3e-4$ | - | - | $3e-4$ | $3e-4$ |
| **w** Learning rate | $3e-4$ | - | - | $3e-4$ | - |
| $f$ Learning rate | - | $5e-5$ | $1e-5$ | - | - |
| $w$ Learning rate | - | $5e-5$ | $1e-5$ | - | - |
| $u$ Learning rate | - | - | $1e-2$ | - | - |
| $\psi^\pi$, $Q^\pi$ Mini-batch size | 256 | - | - | 256 | 256 |
| **w**, $f$, $w$, $u$, Mini-batch size | 2048 | 2048 | 2048 | 2048 | - |
| $\psi^\pi$, $Q^\pi$ Target update rate | 0.005 | - | - | 0.005 | 0.005 |

**Visualizations.** We graph the log MSE between the estimate of $R(\pi)$ and the true $R(\pi)$, where the log MSE is computed as $\log 0.5(X - R(\pi))^2$. We smooth the learning curves over a uniform window of 10. Agents were evaluated every 1k time steps and performance is measured over 250k time steps total. Markers are displayed every 25k time steps with offset for visual clarity.

### F.2 ATARI

We interface with Atari by OpenAI gym (version 0.17.2) (Brockman et al., 2016), all agents use the NoFrameskip-v0 environments that include sticky actions with $p = 0.25$ (Machado et al., 2018b).

**Pre-processing.** We use standard pre-processing steps based on Machado et al. (2018b) and Castro et al. (2018). We base our description on (Fujimoto et al., 2019a), which our code is closely based on. We define the following:

- Frame: output from the Arcade Learning Environment.
- State: conventional notion of a state in a MDP.
- Input: input to the network.

The standard pre-processing steps are as follows:

- Frame: gray-scaled and reduced to $84 \times 84$ pixels, tensor with shape $(1, 84, 84)$.
- State: the maximum pixel value over the 2 most recent frames, tensor with shape $(1, 84, 84)$.
- Input: concatenation over the previous 4 states, tensor with shape $(4, 84, 84)$.

The notion of time steps is applied to states, rather than frames, and functionally, the concept of frames can be abstracted away once pre-processing has been applied to the environment.

The agent receives a state every 4th frame and selects one action, which is repeated for the following 4 frames. If the environment terminates within these 4 frames, the state received will be the last 2 frames before termination. For the first 3 time steps of an episode, the input, which considers the

previous 4 states, sets the non-existent states to all 0s. An episode terminates after the game itself terminates, corresponding to multiple lives lost (which itself is game-dependent), or after 27k time steps (108k frames or 30 minutes in real time). Rewards are clipped to be within a range of $[-1, 1]$.

Sticky actions are applied to the environment (Machado et al., 2018b), where the action $a_t$ taken at time step $t$, is set to the previously taken action $a_{t-1}$ with $p = 0.25$, regardless of the action selected by the agent. Note this replacement is abstracted away from the agent and data set. In other words, if the agent selects action $a$ at state $s$, the transition stored will contain $(s, a)$, regardless if $a$ is replaced by the previously taken action.

**Experiments.** For the main experiments we use a behavior and target policy derived from a Double DQN agent (Van Hasselt et al., 2016), a commonly used deep reinforcement learning algorithm. The behavior policy is an $\epsilon$-greedy policy with $\epsilon = 0.1$ and the target policy is the greedy policy (i.e. $\epsilon = 0$). In Section C.5 we perform two additional experiments with a different behavior policy. Otherwise, all hyper-parameters are fixed across experiments. For each, the data set contains 1 million transitions and uses a discount factor of $\gamma = 0.99$. Each experiment is evaluated over 3 seeds.

**Pre-training.** Both SR-DICE and Direct-SR rely on pre-training the encoder-decoder and deep successor representation $\psi$. Similar to the continuous-action tasks, these networks were trained for 30k and 100k time steps respectively.

**Architecture.** We use the same architecture as most value-based deep reinforcement learning algorithms for Atari, e.g. (Mnih et al., 2015; Van Hasselt et al., 2016; Schaul et al., 2016). This architecture is used for all networks, other than the encoder-decoder network, for fair comparison and was not tuned in any way.

The network has a 3-layer convolutional neural network (CNN) followed by a fully connected network with a single hidden layer. As mentioned in pre-processing, the input to the network is a tensor with shape $(4, 84, 84)$. The first layer of the CNN has a kernel depth of 32 of size $8 \times 8$ and a stride of 4. The second layer has a kernel depth of 32 of size $4 \times 4$ and a stride of 2. The third layer has a kernel depth of 64 of size $3 \times 3$ and a stride of 1. The output of the CNN is flattened to a vector of 3136 before being passed to the fully connected network. The fully connected network has a single hidden layer of 512. Each layer, other than the output layer, is followed by a ReLU activation function. The final layer of the network outputs $|\mathcal{A}|$ values where $|\mathcal{A}|$ is the number of actions.

The encoder-decoder used by SR-DICE and SR-Direct has a slightly different architecture. The encoder is identical to the aforementioned architecture, except the final layer outputs the feature vector $\phi(s)$ with 256 dimensions and is followed by a ReLU activation function. The next state decoder uses a single fully connected layer which transforms the vector of 256 to 3136 and then is passed through three transposed convolutional layers each mirroring the CNN. Hence, the first layer has a kernel depth of 64, kernel size of $3 \times 3$ and a stride of 1. The second layer has a kernel depth of 32, kernel size of $4 \times 4$ and a stride of 2. The final layer has a kernel depth of 32, kernel size of $8 \times 8$ and a stride of 4. This maps to a $(1, 84, 84)$ tensor. All layers other than the final layer are followed by ReLU activation functions. Although the input uses a history of the four previous states, as mentioned in the pre-processing section, we only reconstruct the succeeding state without history. We do this because there is overlap in the history of the current input and the input corresponding to the next time step. The reward decoder is a linear function without biases.

**Network hyper-parameters.** Our hyper-parameter choices are based on standard hyper-parameters based largely on (Castro et al., 2018). All networks are trained with the Adam optimizer (Kingma & Ba, 2014). We use a learning rate of $6.25e-5$. Although not traditionally though of has a hyper-parameter, in accordance to prior work, we modify $\epsilon$ used by Adam to be $1.5e-4$. For $\mathbf{w}$ we use a learning rate of $3e-4$ with the default setting of $\epsilon = 1e-8$. For $u$ we use $1e-3$. We use a mini-batch size of 32 for all networks. SR-DICE, Direct-SR and Deep TD update the target network every 8k time steps. All hyper-parameters are described in Table 3.

**Visualizations.** We use identical visualizations to the continuous-action environments. Graphs display the log MSE between the estimate of $R(\pi)$ and the true $R(\pi)$ of the target policy, where the log MSE is computed as $\log 0.5(X - R(\pi))^2$. We smooth the learning curves over a uniform window of 10. Agents were evaluated every 1k time steps and performance is measured over 250k time steps total. Markers are displayed every 25k time steps with offset for visual clarity.

Table 3: Training hyper-parameters for the Atari domain.

| Hyper-parameter | SR-DICE | DualDICE | GradientDICE | Direct-SR | Deep TD |
|---|---|---|---|---|---|
| Optimizer | Adam | Adam | Adam | Adam | Adam |
| $\psi^\pi$, $Q^\pi$ Learning rate | $6.25e-5$ | - | - | $6.25e-5$ | $6.25e-5$ |
| $\psi^\pi$, $Q^\pi$, $f$, $w$ Adam $\epsilon$ | 1.5e$-$4 | 1.5e$-$4 | 1.5e$-$4 | 1.5e$-$4 | 1.5e$-$4 |
| $\mathbf{w}$, $u$ Adam $\epsilon$ | 1e$-$8 | - | 1e$-$8 | 1e$-$8 | - |
| $\mathbf{w}$ Learning rate | $3e-4$ | - | - | $3e-4$ | - |
| $f$ Learning rate | - | $6.25e-5$ | $6.25e-5$ | - | - |
| $w$ Learning rate | - | $6.25e-5$ | $6.25e-5$ | - | - |
| $u$ Learning rate | - | - | $1e-3$ | - | - |
| $\psi^\pi$, $Q^\pi$ Mini-batch size | 32 | - | - | 32 | 32 |
| $\mathbf{w}$, $f$, $w$, $u$, Mini-batch size | 32 | 32 | 32 | 32 | - |
| $\psi^\pi$, $Q^\pi$ Target update rate | 8k | - | - | 8k | 8k |

