# OpenReview forum: "Practical Marginalized Importance Sampling with the Successor Representation"
_ICLR.cc/2021/Conference — Reject_

### Official Review · AnonReviewer2 · 2020-10-18
**More experiments are necessary to understand how and why SR-DICE works**

**Rating:** 6
**Confidence:** 5

**Review:**

The authors propose SR-DICE based on deep SR for density ratio learning. Empirical advantages are observed in tested domains. Overall I think the idea is interesting and theoretically sound, but the experiments are not fully convincing.

It looks the main claim is that SR-DICE is better than other MIS methods because SR-DICE delegates the update propagation over the MDP to SR, while other MIS methods consider update propagation and density ratio learning together. To me this claim is not coupled with function approximation at all, so I would like to first see some experiments in the tabular setting. SR-DICE is a two-stage learning algorithm, i.e., SR learning + density ratio learning, both have hyperparameters to be tuned. GradientDICE and DualDICE are one-state learning algorithm. If in the tabular setting, we can empirically verify that under the best hyperparameter configuration of each algorithm (guaranteed by a thorough grid search), SR-DICE is more data efficient (counting the samples used in both stages) than GradientDICE and DualDICE, in terms of the density ratio prediction error, then the argument can be well backed. Well-controlled experiments like this, however, do not appear in the current submission.

Once deep networks are used for function approximation, we run into the problem of representation learning. The authors should at least include one more experiment, where MIS methods run directly on the pretrained deep SR features \psi_\pi(s) and/or \phi(s). In this way, we can distinguish whether the empirical advantage of SR-DICE comes from SR-DICE itself or the improved representation learning.

I'm also interested in seeing experiments for larger gammas, e.g., 0.999, 0.9999. I'm wondering if SR-DICE can consistently outperform GradientDICE with increasing discount factors.

Overall, I'm happy to increase the score if I have any misunderstanding or more convincing results are presented. I appreciate that the authors include deep TD and behavior R(\pi) as baselines. The empirical study has independent interest beyond SR-DICE. Moreover, deep TD is also referred to as Fitted-Q-Evaluation (FQE) in [x].

x: Voloshin, Cameron, et al. "Empirical Study of Off-Policy Policy Evaluation for Reinforcement Learning." arXiv preprint arXiv:1911.06854 (2019).

=======================

(Nov 24) The author response addressed my concerns and I therefore raised my score from 5 to 6. I particularly like the idea of using successor representation for density ratio learning.

---

> ### Author Response · Authors · 2020-11-19
> **Response to R2**
>
> Thank you for your review. We have added several of your suggested experiments and hope to add more, we hope this addresses some of your concerns. The main thing we would like to clear up is that the implication that we are uncertain why SR-DICE works is misleading. We have a pretty good idea why SR-DICE achieves a high performance. Although the theory for deep RL is somewhat lacking (for the field itself), we know that, empirically, it works well and thus we have a strong method for estimating the SR. Given the SR we have derived a simple convex loss function on a linear function, so it is unsurprising that this linear function achieves the desired ratio, given a good estimate of the SR. We have a two-stage algorithm where neither stage is particularly complicated and the first stage (learning the SR) has a healthy selection of prior work to build on.
>
> However, what is more uncertain is why this necessarily outperforms the prior MIS methods. Here we have a few ideas as mentioned in the paper. Prior methods use minimax optimization which is more difficult than our optimization problem. Additionally, in 3.2 (previously 4.2) we also draw connections between DualDICE and residual learning, which, historically, performs worse than deep RL. While we certainly don't disagree that this question deserves additional exploration, we would argue that the onus of diagnosing the shortcomings of prior methods should not be entirely on the authors introducing a different approach.
>
> > It looks the main claim is that SR-DICE is better than other MIS methods because SR-DICE delegates the update propagation over the MDP to SR, while other MIS methods consider update propagation and density ratio learning together. To me this claim is not coupled with function approximation at all, so I would like to first see some experiments in the tabular setting.
>
> In the tabular setting, the SR, as well as DualDICE, can be solved directly from matrix operations. For SR-DICE the density ratio has a closed form solution from the given SR, which returns the same value estimate as directly using the SR. Consequently, we can say SR-DICE achieves the same performance as TD learning or model-based learning, depending on the chosen method for computing the SR. We have added a short derivation of this result to Appendix B. Consequently, decoupling the update is only interesting when function approximation is included.
>
> That being said, we agree that some simplified experiments would improve the paper and we aim to include these by the end of the rebuttal phase. *** Nov 24 update: we have added this experiment to the Appendix (C.3, Figure 9). Our results show there is still an empirical benefit of SR-DICE over the baseline MIS methods even in a very simple 5-state MDP domain. We hope this addresses your concern.
>
> > SR-DICE is a two-stage learning algorithm, i.e., SR learning + density ratio learning, both have hyperparameters to be tuned. GradientDICE and DualDICE are one-state learning algorithm.
>
> Clarification: each stage uses the same data so the fact that SR-DICE is a two-stage process only matters from a computational aspect, rather than a sample-efficiency one. From the computational side-- in the appendix we show the run time of SR-DICE is less than GradientDICE and DualDICE.
>
> > Once deep networks are used for function approximation, we run into the problem of representation learning. The authors should at least include one more experiment, where MIS methods run directly on the pretrained deep SR features \psi_\pi(s) and/or \phi(s). In this way, we can distinguish whether the empirical advantage of SR-DICE comes from SR-DICE itself or the improved representation learning.
>
> This is a great idea, thank you. This experiment has been added (Figure 7) to Appendix C.2. Generally, we find that the representation does not improve the performance of DualDICE or GradientDICE, except in the instance where DualDICE is made linear and uses the $\psi^\pi$ representation. However, even in this best case, it still underperforms SR-DICE and exhibits a large variance.
>
> > I'm also interested in seeing experiments for larger gammas, e.g., 0.999, 0.9999. I'm wondering if SR-DICE can consistently outperform GradientDICE with increasing discount factors.
>
> We have added 0.999 to Figure 5. SR-DICE and GradientDICE perform roughly the same but neither perform well. This is not surprising as the deep RL component (naively) struggles with large gammas.

---

### Official Review · AnonReviewer3 · 2020-10-26
**Not convinced by both the theoretical and empirical results**

**Rating:** 6
**Confidence:** 4

**Review:**

The paper proposes SR-DICE, which uses a successor representation to compute DICE (discounted stationary distribution correction term).
* I am worried about both the technical and experimental qualities of this work. The theorems presented are either obvious or previously presented in other works. While the authors argue that the marginalized importance ratio is independent of the horizon (I assume that they are talking about the variance), MIS only alleviates the estimator variance's exponential dependence on the horizon to become the polynomial dependence on the horizon (as proved in Xie et al., Towards Optimal Off-Policy Evaluation for Reinforcement Learning with Marginalized Importance Sampling, 2019). In the experiments, it is hard to believe that the GradientDICE and DualDICE perform that poorly having Log MSE larger than 0 while the GenDICE paper reports Log MSE less than -4 (HalfCheetah).
* The paper uses $\phi$ and $\psi$ learned by the previous deep successor representation learning algorithm, which is not meant to be used to learn marginal importance ratio. In particular, $\phi$ is learned by minimizing state, action, and reward reconstruction error and $\psi$ is the discounted sum of $\phi$. If we consider a case where $\pi$ only exploits a very small subset of state-action space, it is easy to see that the reconstruction error minimization in the dataset is not an optimal representation for the marginal importance ratio learning. In this sense, only the linear vector $w$ is used for the learning of marginal importance ratio.
* The experiment setting is not fair. Direct-SR and SR-DICE in their implementation have effectively 2 hidden layers, where DualDICE and GradientDICE in their implementation have a single hidden layer.
* The paper is hard to follow. Especially, notation abuse between the real reward and the virtual reward which is optimized to give a marginal importance ratio is very confusing (abuse between real Q and minimizer Q as well). Section 4.2 is also confusing because the authors imposes the problem of DualDICE that is not actually handled by SR-DICE.
* The idea of adopting successor representation for learning marginal importance ratio seems quite novel.
* Some people will be interested in this work, but I think the paper would not have much impact on the field.

Overall,

PROS:
* The idea of using successor representation for learning marginal importance ratio is novel.
* Avoids minimax formulation of other DICE algorithms, which makes the optimization very hard.

CONS:
* Not very meaningful theoretical results are presented, which mostly just confuse readers.
* Uses the representation that is not learned for marginal importance ratio learning
* Questionable experiment results


Minor details:
* y axis label is "Log MSE" for figures although the y axis is log scaled MSE.


--------------------------------------
Most of the concerns are addressed by the authors, and I raised my score accordingly.

---

> ### Author Response · Authors · 2020-11-19
> **Response to R3**
>
> Thank you for your review. We have corrected the presentation of the empirical results, added experiments to the appendix, and tried to improve clarity. We address your comments below.
>
> > While the authors argue that the marginalized importance ratio is independent of the horizon (I assume that they are talking about the variance), MIS only alleviates the estimator variance's exponential dependence on the horizon to become the polynomial dependence on the horizon [...]
>
> The MIS ratio is independent of horizon by definition, as it is the ratio of the occupancy of state-action pairs, in comparison to traditional IS where the ratio is composed of a product over a trajectory. The learning procedure of MIS is still dependent on horizon. We state in several places that this results in a reduction in variance, and in the first two sentences of Section 3.2 (previously Section 4.2) explicitly state “One of the main attractions for MIS methods is they use importance sampling ratios which are independent of horizon. While factually correct, we remark the optimization problem is not independent of horizon and MIS methods are still subject to the dynamics of the underlying MDP.” That being said, the discussion of the MIS ratio being independent of horizon only appears in motivating statements/introduction, so it is unclear why this is mentioned as a major detail of our work.
>
> > In the experiments, it is hard to believe that the GradientDICE and DualDICE perform that poorly having Log MSE larger than 0 while the GenDICE paper reports Log MSE less than -4 (HalfCheetah).
>
> This is a good catch. We had two issues. Firstly we used a $\log_{10}$ scale rather than the typical $\log_e$ scale. Secondly, we measured error as $(x - y)^2$ rather than $0.5 (x - y)^2$ as previous papers have done. We added an additional experiment (Figure 2) which mimics the experimental setup as GenDICE (or as close as we could, given their GitHub is empty & some experimental details are unmentioned in the paper). We also performed additional hyper-parameter searching to improve the DualDICE baseline. We find that our GradientDICE implementation matches the log MSE of -4 (or less) on HalfCheetah while our SR-DICE achieves around -7.5. DualDICE falls short on HalfCheetah but reaches low error on several other environments.
>
> > If we consider a case where $\pi$ only exploits a very small subset of state-action space, it is easy to see that the reconstruction error minimization in the dataset is not an optimal representation for the marginal importance ratio learning. In this sense, only the linear vector w is used for the learning of marginal importance ratio.
>
> We consider only using the linear vector a strength of the method as learning a linear vector is far easier than a deep network and we maintain the representation benefits from a neural network by training the SR separately. How $\phi$ is learned is a design choice which we have based on previous SR methods but could be easily modified. However, our empirical results demonstrate that this is a reasonable choice. Consider also that since MIS is only applied to $(s,a) \in \mathcal{D}$, the representation does not necessarily need to account for $(s,a) \notin \mathcal{D}$.
>
> > The experiment setting is not fair. Direct-SR and SR-DICE in their implementation have effectively 2 hidden layers, where DualDICE and GradientDICE in their implementation have a single hidden layer.
>
> We have added an experiment (Figure 7) to Appendix C.2. which passes different representations ($\phi$ and $\psi^\pi$) to DualDICE and GradientDICE, and also an experiment (Figure 8) which provides DualDICE and GradientDICE an additional hidden layer. We do not find that increasing the network capacity of these previous methods improves performance.
>
> > The paper is hard to follow. Especially, notation abuse between the real reward and the virtual reward which is optimized to give a marginal importance ratio is very confusing (abuse between real Q and minimizer Q as well).
>
> We have added several clarifying statements to Section 4 and denoted the virtual reward as $\hat r$ and the real reward as $r$. Thank you for pointing out this issue.
>
> > Some people will be interested in this work, but I think the paper would not have much impact on the field.
>
> See our general response for our thoughts on impact.
>
> > y axis label is "Log MSE" for figures although the y axis is log scaled MSE.
>
> As noted above this has been corrected, thank you.

---

> > ### Comment · AnonReviewer3 · 2020-11-20
> > **About the first point**
> >
> > I raised my score as the authors addressed most of my concerns. However, I am not sure whether we should consider the statement "A key claim of MIS methods is they reduce the variance ... by using ratios which are independent of trajectory length or horizon" valid. Is there any reference that claims this as well? While authors argue that it is factually correct, I disagree, because the underlying optimal ratio is dependent on the effective horizon (gamma), and the estimated ratio will approach the underlying optimal ratio as we gather more diverse data, use a more flexible network, and optimize carefully. For me, it sounds similar to saying: "Using critics instead of empirical return is advantageous because the critic estimators are independent of the horizon and empirical return is dependent on the horizon."

---

> > > ### Author Response · Authors · 2020-11-20
> > > **You are correct, we have now addressed this**
> > >
> > > Thank you for the quick response and adjusting your review.
> > >
> > > An important paper in this area which makes this claim is [1], which we based these motivating statements on (example quote):
> > > > In this work, we develop a new approach that tackles the curse of horizon. The key idea is to apply importance sampling on the average visitation distribution of single steps of state-action pairs, instead of the much higher dimensional distribution of whole trajectories. This avoids the cumulative product across time in the density ratio, substantially decreasing its variance and eliminating the estimator’s dependence on the horizon.
> > >
> > > However, as you mentioned we now know the variance is still dependent on the horizon [2,3], so the above quote is arguably overstated. Consequently, I will backpedal and admit you are correct. I will re-affirm that we certainly never claimed that the variance was independent of the horizon, but the statement that the ratios are independent of the horizon is easily misunderstood. MIS ratios are independent of horizon in the sense that they do not need to consider the length of trajectories of the behavior policy, however they are not independent of the horizon defined implicitly by the discount factor (as $d^\pi$ is defined as a function of $\gamma$). This is clearly a problematic confusion point, and we have corrected it in the current iteration of the paper. Thank you for continuing this discussion.
> > >
> > > References
> > > - [1] Qiang Liu, Lihong Li, Ziyang Tang, and Dengyong Zhou. Breaking the curse of horizon: Infinite-horizon off-policy estimation. 2018.
> > > - [2] Tengyang Xie, Yifei Ma, and Yu-Xiang Wang. Towards optimal off-policy evaluation for reinforcement learning with marginalized importance sampling. 2019.
> > > - [3] Yao Liu, Pierre-Luc Bacon, and Emma Brunskill. Understanding the curse of horizon in off-policy evaluation via conditional importance sampling. 2019.

---

### Official Review · AnonReviewer1 · 2020-10-26
**Practical Marginalized Importance Sampling with the Successor Representation**

**Rating:** 6
**Confidence:** 3

**Review:**

***Summary***
The paper proposes an approach to employ successor representation combined with marginalized importance sampling. The basic idea exploited in the paper consists of expressing the occupancies in terms of the successor representation and to model it via a linear combination of some features. This allows handling, although approximately, continuous state-action spaces. After having derived the objective function, an experimental evaluation on both Mujoco and Atari domains is presented, including an ablation study.

***Major***
- (About the linearity of the weight) Linear representations expressed in terms of a feature function are common in RL as the reward function can be often seen as a trade-off of different objectives encoded in the features. However, the choice of the linear representation in Equation (7) is based on the assumption that the marginalized weight is linear in the feature function. This assumption seems to me less justified compared to the one for the reward function. Clearly, a suitable feature design could overcome this limitation. Can the authors explain how the features \phi are selected or learned?
- (Experimental evaluation) The results presented in the experimental evaluation are partially unsatisfactory, as also the authors acknowledge. It seems that there is no clear benefit in employing the marginalized importances sampling (both the baselines and the proposed approach) compared to standard deep temporal difference approaches. The authors suggest that this phenomenon can be ascribed to the fact that the quality of the marginalized weights is affected by the successor representation learned. I don't think this is the main weakness of the paper, but a reflection of the usefulness of the method in complex scenarios is necessary. Alternatively, it would be interesting to compare the proposed approach with DualDICE and GradientDICE on simpler tasks (maybe toy ones) in which DualDICE and GradientDICE work well.


***Minor***
- The related work section should be moved later in the paper, maybe after Section 4
- Pag 2, two lines above Equation (2): the transition model is here employed as a distribution over the next state s' and the reward r, but the reward function is considered separately in the definition of MDP presented before
- Figures 2, 3, and 4: the plots are not readable when printing the paper in grayscale. I suggest using different linestyles and/or markers

***Typos***
- Pag 2: isn't -> is not
- Pag 2: doesn't -> does not
- Pag 8: the the -> the

***Overall***
The paper can be considered incremental compared to DualDICE. I did not find any fault, but I feel that the significance of contribution is currently insufficient for publication at ICLR. In particular, for a paper that proposes a practical variation of a theoretically sound algorithm, the experimental evaluation is essential. I think that the results are currently unable to clearly show the advantages of the proposed method.

---

> ### Author Response · Authors · 2020-11-19
> **Response to R1**
>
> Thank you for your review. We have made several adjustments to the paper based on your suggestions and discuss some of the finer points below.
>
> > Can the authors explain how the features \phi are selected or learned?
>
> This is explained in detail in Appendix D, but we have added some explanation to the main body in Section 3.1, along with the algorithm pseudocode. The features $\phi(s,a)$ are learned via an encoder-decoder which reconstructs the transition. This choice is based on prior methods [1,2], and by our experimental results, we can see it works well.
>
> > The results presented in the experimental evaluation are partially unsatisfactory, as also the authors acknowledge. It seems that there is no clear benefit in employing the marginalized importances sampling (both the baselines and the proposed approach) compared to standard deep temporal difference approaches.
>
> We consider this a strength of the paper. Previous MIS papers omit deep TD approaches and so our more extensive empirical evaluation provides commentary on the area of MIS as a whole. As we mention in the general response, just because we are not SOTA at OPE, our large improvements over previous MIS methods should still provide value to anyone working in this area. Additionally, as mentioned in both the paper and our general response, the ratios themselves have also been shown to be useful in a number of settings and our improvements could be applied to these settings as well.
>
> > Alternatively, it would be interesting to compare the proposed approach with DualDICE and GradientDICE on simpler tasks (maybe toy ones) in which DualDICE and GradientDICE work well.
>
> We agree. There are experiments on Reacher and Pendulum, environments contained in the DualDICE and GradientDICE paper, found in Appendix C.1 (Figure 6) but we will try to include a more extensive evaluation on simpler tasks before the end of the rebuttal phase. *** Nov 24 update: we have added some experimental results on a simple toy task with varied feature representations (Appendix C.3, Figure 9).
>
> > The related work section should be moved later in the paper, maybe after Section 4
>
> Done. Thank you for the suggestion.
>
> > Figures 2, 3, and 4: the plots are not readable when printing the paper in grayscale. I suggest using different linestyles and/or markers
>
> Thank you for the suggestion. All figures have been reformatted with markers.
>
> > The experimental evaluation is essential. I think that the results are currently unable to clearly show the advantages of the proposed method.
>
> We believe the experimental evaluation clearly shows a large benefit of our method over previous MIS methods. We believe that improving in this subfield is an interesting direction for OPE, even if we also show it is not SOTA. We have chosen to include deep RL results to give a more comprehensive view of OPE but it seems unfair to say they invalidate our results.
>
> References
> - [1] Tejas D Kulkarni, Ardavan Saeedi, Simanta Gautam, and Samuel J Gershman. Deep successor reinforcement learning. 2016.
> - [2] Marlos C Machado, Marc G Bellemare, and Michael Bowling. Count-based exploration with the successor representation. 2018.

---

> > ### Comment · AnonReviewer1 · 2020-11-24
> > **Response to Authors**
> >
> > I thank the authors for the detailed feedback. I appreciate the adjustments to the experimental evaluation part. Thus, I am raising my score to 6.

---

### Official Review · AnonReviewer4 · 2020-10-28
**Theorems are vacuous, and do not seem to correspond to what the experiments demonstrate.**

**Rating:** 5
**Confidence:** 4

**Review:**


Strengths:

1) The experiments are extensive, and clearly demonstrate the merits as compared to prior benchmarks for off policy RL.

2) The contextual discussion is clear, well-motivates the proposed approach, and gives a nice overview of how importance sampling and off policy RL intersect.



Weaknesses:


1) Theorem 1 seems vacuous. The proof is a simple exercise in elementary calculus -- one may easily show the minimizer of a quadratic is the least squares estimator. The authors need to better explain what is the technical novelty of this statement, and why it is meaningful. Upon inspection it does not seem to qualify as a theorem. This is also true of Theorems 2 and 3. Therefore, I feel the conceptual contribution is not enough to warrant acceptance.

2) The notion of successor representation seems identical to the occupancy measure, which in order to estimate, requires density estimation, which is extremely sample inefficient. Can the authors comment about how to estimate the successor representation efficiently? There is very little discussion of sample complexity throughout, which is somewhat alarming because a key selling point of off-policy schemes for RL is that they alleviate the need for sampling from the MDP transition dynamics.

3) The actual algorithm pseudo-code is missing from the body of the paper, which is permissible because it is in the appendix. However, the structural details of how the algorithm works iteratively and how it departs from previous works are also not explained. That is, while derivation details are presented, iterative details are not. In my opinion this should be strictly required in the body of the paper, as well as contextual discussion of what is similar/different from previous works, but all I could find was high level presentation of objectives minimized at various sections, but not how they are interlinked.

4) The background discussion is disjointed. There is a preliminaries section on page 5, as well as a background section 3.


Minor Comments:


1) References missing related to the sample/parameterization complexity issues associated with importance sampling:

Koppel, A., Bedi, A. S., Elvira, V., & Sadler, B. M. (2019). Approximate shannon sampling in importance sampling: Nearly consistent finite particle estimates. arXiv preprint arXiv:1909.10279.

---

> ### Author Response · Authors · 2020-11-19
> **Response to R4**
>
> Thank you for your review. We address each of your concerns below and hope you feel some of the changes made will strengthen the paper.
>
> > Theorem 1 seems vacuous. [...] The authors need to better explain what is the technical novelty of this statement, and why it is meaningful. [...] This is also true of Theorems 2 and 3.
>
> As stated in the paper, Theorem 1 is not a novel result and is derived directly from the DualDICE paper. We have re-labeled Theorem 1 as Observation 1. The objective of the theorems are not to provide deep technical novelty but provide a more rigorous justification to the algorithm. We argue that while Observation 1 and Theorem 1 (previously Theorem 2) may not be interesting to an experienced reader, they are necessary and beneficial for anyone less familiar with the material. Theorem 2 (previously Theorem 3) draws an interesting connection between DualDICE and reward functions and allows us to better understand the optimization problem defined by DualDICE.
>
> > The notion of successor representation seems identical to the occupancy measure, which in order to estimate, requires density estimation, which is extremely sample inefficient. Can the authors comment about how to estimate the successor representation efficiently?
>
> The SR can be learned efficiently through TD learning. This is important in the context of practical problems, which require deep networks-- since there are solid deep RL methods for TD learning. We explain the practical aspect of this computation in the background as well as Appendix D.
>
> > There is very little discussion of sample complexity throughout, which is somewhat alarming because a key selling point of off-policy schemes for RL is that they alleviate the need for sampling from the MDP transition dynamics.
>
> We have experiments on sample complexity in Figure 5 (a). The sample efficiency of SR-DICE is identical to the sample efficiency of learning the SR since optimizing the reward function/$\mathbf{w}$ requires no additional samples.
>
> > The actual algorithm pseudo-code is missing from the body of the paper, which is permissible because it is in the appendix.
>
> The algorithm has been added to Section 3.1. Thank you for this concrete suggestion.
>
> > The background discussion is disjointed. There is a preliminaries section on page 5, as well as a background section 3.
>
> We have merged the DualDICE preliminaries into the background section. Thank you.

---

### Author Response · Authors · 2020-11-19
**General response to reviewers**

Thank you to the reviewers for their time, comments and for providing concrete suggestions on how to improve the paper.

We would like to begin by clarifying the objective and impact of our paper. The objective of this paper was to scale a subfield of OPE methods, MIS algorithms, to high dimensional problems. We believe we have succeeded overwhelmingly at this objective and our proposed algorithm greatly outperforms previous MIS algorithms over a wide range of domains and data sets.

We believe that OPE is an area grounded in an empirical objective (evaluating policies) and consequently, there is great value in scaling algorithms to practical, high dimensional tasks. A consequence of our empirical evaluation is we learn that, although our approach is competitive, deep RL algorithms still outperform our method (and MIS methods) on these high dimensional tasks. However, we believe that:
1. There is space for algorithms which improve in their subfield but fall short of SOTA (example: see any ML algorithm which is not a neural network)
2. MIS methods are a promising research direction for reasons other than OPE (example: applications of MIS ratios [1,2,3,4]).
3. Showing current MIS methods underperform deep RL is of independent interest beyond our proposed algorithm (as mentioned by R2).

This paper does not:
- Provide deep theoretical insights on MIS methods.
- Improve SOTA for OPE.

This paper **does**:
- Provide a simple MIS method which is competitive with deep RL algorithms and greatly outperforms previous MIS methods.
- Provide a comprehensive empirical evaluation of current MIS methods on challenging high dimensional benchmarks.
- Provide rigorous reproduction details including code, transparent baselines omitted by previous papers (deep RL, behavior policies), run time analysis, and exact experimental details.

We hope the reviewers consider recalibrating their expectations of what constitutes a good OPE paper. Of course, the paper is not perfect, and we've done our best to include as many of the suggestions from the reviewers as possible, extending the paper from eight to nine pages + more in the appendix.

Major Changes:
+ New experimental results for our 7 continuous control environments with a new setting matching GenDICE (Figure 2). (R3)
+ New experimental results examining the impact of the representation (both $\phi$ and $\psi$) on previous MIS methods (Appendix C.2, Figure 7). (R2)
+ New experimental results providing previous MIS methods larger networks (Appendix C.2, Figure 8). (R3)
+ Expanded the ablation on the discount factor to include 0.999 (Figure 5). (R2)
+ Included discussion on the tabular variant of SR-DICE (Appendix B). (R2)
+ New experimental results on a simple toy domain, with hyper-parameter fine-tuning (Appendix C.3, Figure 9) (R1, R2) *** Added Nov 24

Minor Changes:
+ Moved algorithm description from appendix to main body and included additional algorithmic details. (R4)
+ Re-did every graph to include markers for visual clarity and fixed the log scale. (R1, R3)
+ Performed additional hyper-parameter tuning on the DualDICE baseline. (R3)
+ Merged DualDICE preliminaries into background section. (R4)
+ Moved related work after the main body. (R1)
+ Adjusted the notation on the learned reward to provide better distinction from the environment reward. (R3)
+ Corrected typos and other minor presentation issues. (R1, R3)

TODO:
+ R1,R2 both suggested performing experiments on simpler settings. To some extent this is tangential to the objective of the paper (scaling MIS to more challenging domains) but we agree that some interesting insight could be derived toy, or comparatively easier, domains. We will attempt to add this to the paper before the end of the rebuttal phase. *** Completed Nov 24

References:
- [1] Yao Liu, Adith Swaminathan, Alekh Agarwal, and Emma Brunskill. Off-policy policy gradient with state distribution correction. 2019.
- [2] Ofir Nachum, Bo Dai, Ilya Kostrikov, Yinlam Chow, Lihong Li, and Dale Schuurmans. Algaedice: Policy gradient from arbitrary experience. 2019.
- [3] Ilya Kostrikov, Ofir Nachum, and Jonathan Tompson. Imitation learning via off-policy distribution matching. 2019.
- [4] Ahmed Touati, Amy Zhang, Joelle Pineau, and Pascal Vincent. Stable policy optimization via off-policy divergence regularization. 2020.

---

> ### Author Response · Authors · 2020-11-24
> **As we near the end of the discussion phase**
>
> Since most of the reviewers have not yet responded to our rebuttal (understandably, given the time frame and that many reviewers are authors themselves with their own deadlines), we would like to remark that we have made many additions to the paper and believe we have addressed every comment/request made by the reviewers. We hope that this builds some good faith, that if the paper were to be accepted, any additional requests or comments that occur after the discussion phase will still make their way into the final draft.
>
> Thank you again for your time, comments, and contributions towards improving this paper.

---

### Decision · Program_Chairs · 2021-01-07
**Final Decision**

**Decision:**

Reject

**Comment:**

The paper is about an approach that combines successor representation with marginalized importance sampling.
Although the reviewers acknowledge that the paper has some merits (interesting idea, good discussion, extensive experimental analysis) and the authors' responses have solved most of the reviewers' issues, the paper is borderline and the reviewers did not reach a consensus about its acceptance. In particular, the reviewers feel that the contributions of this paper are not significant enough.
I encourage the authors to modify their paper by taking into consideration the suggestions provided by the reviewers and try to submit it to one of the forthcoming machine learning conferences.